# Correlative voltage imaging and cryo-electron tomography bridge neuronal activity and molecular structure

Mingyu Jung [1,7], Gwanho Ko [1,7], Dongsung Lim[1], Seonghoon Kim [2,3,4], Sojeong Kim [5], Young-Joon Kim [5,6], Myunghwan Choi [1,2] & Soung-Hun Roh [1,2]

Neurons exhibit varying electrophysiological properties due to dynamic changes in spatiotemporal molecular networks. In situ cryo-electron tomography (cryo-ET) provides advantages for high-resolution visualization of macromolecular complexes within their cellular context. Although correlation with fluorescent labeling allows cryo-ET to target specific cellular regions, it does not adequately reflect the electrophysiological properties of heterogeneous neurons. To bridge high-resolution molecular imaging with electrophysiological properties of individual neurons, we develop a Correlative Voltage Imaging and cryo-ET (CoVET) technique. The nondestructive nature of voltage imaging is compatible with cryo-ET, enabling a direct correlation between neuronal electrophysiology and molecular structures. Neurons are clustered based on their electrophysiological properties, allowing for single-cell-guided structural analysis using cryo-ET. We analyze the translational landscapes of individual neurons and find distinct structural characteristics and spatial networks among ribosomes from different electrophysiological clusters. Our results highlight the importance of the correlation between the electrophysiological properties and molecular structures.

Various types of neurons in the brain orchestrate functions such as cognition and memory. These neurons exhibit diverse electrophysiological properties defined by their molecular profiles and communicate with each other electrically to mediate network-scale neural computation[1–4]. This electrical communication within neurons triggers the modulation of spatiotemporal molecular networks, such as cytoskeleton remodeling, spatial translation, and proteostasis response[5–8]. The spatial distribution and structural heterogeneity of molecular networks influence neural functions through neural plasticity, ranging from synaptic ultrastructures to network-level connectivity[9,10]. Disruption of these molecular networks results in abnormal neural activity, leading to impaired signal transduction. These disruptions are associated with severe phenotypes, including aging, cognitive decline, and neurodegeneration[11,12]. Thus, there is considerable demand for structural characterization of spatial molecular networks within electrophysiological properties.

To study the molecular networks of neurons, in situ cryo-electron tomography (cryo-ET) is an emerging technique because of its advantages in examining cells and tissues by preserving specimens in their natural state through cryogenic preservation[13,14]. This technique allows for high-resolution visualization of macromolecular complexes

[1]School of Biological Sciences, Seoul National University, Seoul, Republic of Korea. [2]Institute of Molecular Biology and Genetics, Seoul National University, Seoul, Republic of Korea. [3]Institute for Brain and Cognitive Sciences, Tsinghua University, Beijing, China. [4]Hangzhou Zhuoxi Institute of Brain and Intelligence, Hangzhou, China. [5]Department of Semiconductor Engineering, Gachon University, Seongnam, Republic of Korea. [6]Department of Electronic Engineering, Gachon University, Seongnam, Republic of Korea. [7]These authors contributed equally: Mingyu Jung, Gwanho Ko. ✉e-mail: choim@snu.ac.kr; shroh@snu.ac.kr

within a cellular context. Recent studies have highlighted its capabilities, such as visualizing translational landscapes of bacterial ribosomes at 3.5 Å resolution[15] and identifying a uncharacterized microtubule inner protein in sperm through in situ high-resolution imaging with alphafold2[16]. This technique expands its role in biology by enabling the analysis of spatial and temporal molecular events in complex environments, including the dynamics of organelles and proteins.

Cryo-ET is frequently correlated with light microscopy, particularly cryo-correlative light and electron microscopy (cryo-CLEM), to accurately identify regions of interest. Correlation with fluorescent labels is widely used to determine the coordinates of desired proteins or organelles, allowing the corresponding regions to be imaged using cryo-ET[17–19]. This correlated imaging technique has also been adapted to explore neurons by analyzing molecular networks around poly-GA aggregates[18] and the ultrastructural characteristics of excitatory and inhibitory synapses[19]. Although fluorescence labeling is a well-established approach, it cannot accurately represent the electrophysiological properties of individual neurons. This can potentially lead to bias by integrating information from neurons with different electrophysiological properties. Therefore, an additional layer of correlative methods is required to complement electrophysiological information in studying neurons. Voltage imaging is an optical technique that directly characterizes the membrane potential of live neurons by measuring temporal changes in the intensity of voltage-sensitive fluorophores[20–22]. This approach is invaluable for capturing live information on the electrophysiological properties of individual neurons and its nondestructive characteristic is compatible with cryo-ET, allowing for the direct correlation of neuronal function with structures.

In this study, to correlate high-resolution molecular imaging with the electrophysiological properties of individual neurons, we develop a combined approach using voltage imaging and cryo-ET, which we term Correlative Voltage imaging and cryo-ET (CoVET). For the CoVET setup, we design a customized grid holder for the coculture system of neurons and astrocytes, allowing optimization of cultured neurons and efficient correlated imaging. We perform voltage imaging with the application of electric field stimulation to the grid to evoke the synchronous discharge of neurons and vitrified the grids immediately after imaging. The neurons are then classified into three electrophysiological clusters based on their electrophysiological activity. Lamellae are created for each cluster using cryo-focused ion beam/scanning electron microscopy (cryo-FIB/SEM) and image sequentially using cryo-ET (Fig. 1). Via this correlative approach, we show that ribosomes in somas from different electrophysiological clusters exhibit distinct translational landscapes and spatial networks. This emphasizes the need for an electrophysiology-based single-cell cryo-ET to study neurons.

## Results

### Preparing primary neuron for effective CoVET analysis
To demonstrate our combined pipeline, we used rat hippocampal neuron cultures as a reference for primary neuronal culture[23]. Conventional hippocampal primary cultures required a seeding of ~120,000 cells/cm², resulting in a high-density cell population[24]. This high density is beneficial for maintaining neuronal health through the active formation of synapses and exchange of chemicals between neurons. However, high-density cultures lead to clumping of neurons, making it difficult to visually identify single neurons, which is crucial for optical and cryo-ET analysis. To achieve a low-density culture on grids without compromising neuronal health, we employed a sandwich culture[25–28] on grids to optimize the lower cell population. This ensures normal functional development of neurons while maintaining a sufficiently low density to identify individual neurons for both cryo-ET and voltage imaging.

Furthermore, to enable diverse applications of CoVET, we must ensure that voltage imaging on grids is compatible with various optical devices, including live-cell chambers and electric field stimulators. To enhance the compatibility and efficiency of voltage imaging and sandwich culture with CoVET, we devised a doughnut-shaped grid holder made of stainless steel, featuring a central stepped hole with a 0.2 mm depth, to reversibly secure a TEM grid. The 18 mm outer diameter of the grid holder was tailored to be compatible with commercial optical imaging chambers and electric field stimulators. The holder features a central hole, with one face measuring 3.1 mm, larger than the TEM grid, for easy insertion and removal, and the other face measuring 2.5 mm to prevent grid detachment (Fig. 2a). One day prior to seeding the neurons, a grid was mounted onto the grid holder and an 18 mm coverslip was attached to secure the grid. After mounting the grid to the grid holder, vacuum grease was applied using a paint brush at three different spots along the back side of the grid holder (to-be interface between the grid holder and coverslip) to prevent detachment of the coverslip. An 18 mm round coverslip was firmly attached by pressing it against the backside of the grid holder. Gold grids coated with carbon film were glow-discharged and UV-sterilized prior to coating with poly-D-lysine (PDL) and laminin. After sterilization, the grids were packaged into sterilized grid holders and coated with PDL. After washing, grids were coated with laminin for 4 h (Fig. 2b).

In this study, to achieve appropriate spacing between the feeder layer and packaged grid, a commercially available flat-bottomed paraffin wax dot was used. Three paraffin wax dots were slightly melted at the bottom and allowed to attach to the bottom of a 12-well astrocyte feeder layer, which was cultured on the paraffin dots-attached 12-well plate for 2–3 days before neuron culture. On the other hand, ~45,000 cells/cm² were seeded onto the packaged grids. Following 4 h of attachment of the neurons to the grids, the packaged grids were positioned above the precultured astrocyte feeder layer (Fig. 2b). Through low-density culture, the optimal sparsity of neurons (~15000 cells/cm²) on the grids was achieved on the first day in vitro (DIV) and facilitated the correlative observation of neuronal function by optical microscopy and their structures by electron microscopy. This facilitated the correlative observation of neuronal function by optical microscopy and their structures by electron microscopy. To test the efficacy of the sandwich culture on grids, we compared low-density neurons cultured with and without a feeder layer (Fig. 2c). The introduction of the astrocyte feeder layer led to a significant increase in the number of neurites, which was apparent after 4 DIV (Fig. 2c, d). In contrast, neurons without a feeder layer displayed near-complete retraction of neurites by 9 DIV, failing to form a functional neural network (Fig. 2c, d).

### Voltage imaging characterizes and classifies the electric responses of individual neurons on grids
To characterize the electrophysiological properties of individual neurons, cultured cells were pre-incubated with the voltage-sensitive probe BeRST1[29], and the DNA-labeling probe Hoechst was applied concurrently to label the nuclei for further analysis. BeRST1 was selected owing to its excellent brightness, linearity, and response kinetics[29]. The neurons cultured on the grid were secured on the grid holder and then positioned in the imaging chamber with the neurons facing down towards the objective lens of the inverted epifluorescence microscope (Fig. 3a). This arrangement ensured that the imaging of neurons was unobstructed by the grid bars. Two electrodes within the imaging chamber were connected to an electric field stimulator to discharge neurons across the entire grid (Fig. 3b). To validate that our electric field stimulation system with a grid holder can deliver adequate stimulation, we performed electric field simulations (Fig. 3c, Supplementary Fig. 1a, b). Our simulation indicated that, on average, a 1089 V/m electric field was applied to the neurons on the grids, which is sufficient to evoke action potentials[30]. Furthermore, previous study

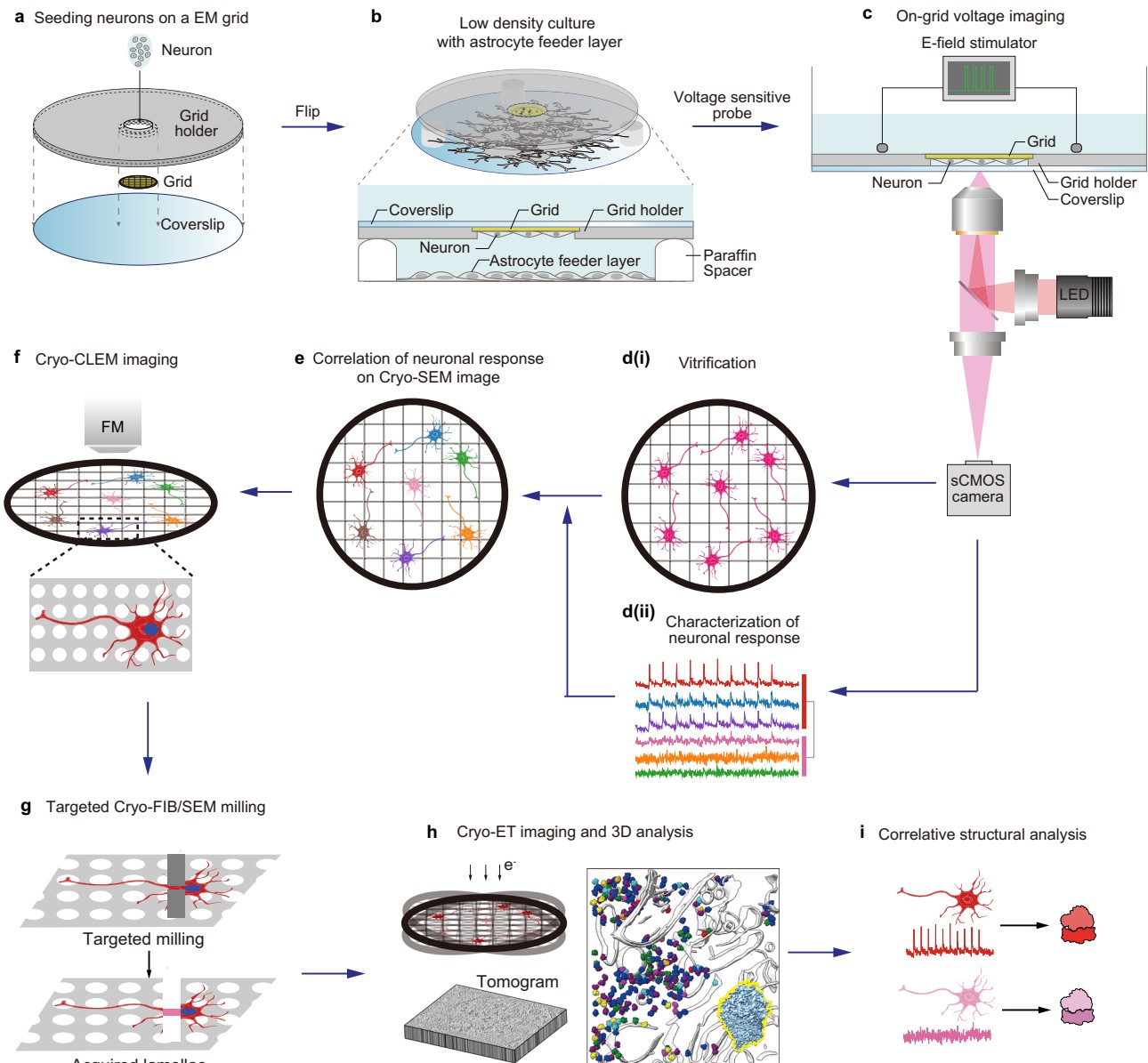

**Fig. 1 | Overview of CoVET. a** Seeding neurons on a grid packaged in a grid holder. **b** Neurons attached to the grid are sandwich cultured with the astrocyte feeder layer. **c** Voltage imaging of neurons on a grid with electric field stimulation. **d(i)** After voltage imaging, grids were vitrified immediately. **d(ii)** Single neuronal responses on the grid were characterized. **e** Neuronal responses were correlated with the Cryo-SEM image. **f** Cryo-CLEM imaging was performed on the grid. **g** Targeted neurons were milled according to their responses. **h** Correlated Cryo-ET imaging and 3D reconstruction were performed. **i** Structural analysis was performed according to neuronal responses.

revealed that isotropic responses across neurons are induced by electric field stimulation in the 2D primary cultured neurons[31]. Based on this information, we validated that our system is capable of evaluating the responses of single neurons on the grids.

We first acquired a montage image of the entire grid to check the distribution of neurons and integrity of the grid (Fig. 3d). After checking the quality of the grids, we performed time-series imaging with ten repetitions of 1 ms electric-field stimulation. We observed that a large population of neurons reliably responded to the electric field stimulation (Fig. 3e). To ensure that the intensity traces of neurons reflected the dynamics of membrane potential, we also tested HEK293T cells, which are electrically non-excitable cells. As expected, electric field stimulation did not evoke any notable changes in intensity traces in HEK293T cells, confirming the validity of our voltage imaging data for neurons on a grid (Supplementary Fig. 2a, b).

To comprehensively characterize the electrophysiological responses of the neurons, we segmented all neurons on the grids and extracted their coordinates along with the voltage traces (Fig. 3d-e). Analysis of these voltage traces revealed a wide range of responses from consistent action potentials to weak, irregular, or completely absent responses (Fig. 3e). To compare heterogeneity among neurons, we measured the peak value, decay parameter, and reproducibility of the action potential after 10 stimulations from the voltage traces of individual neurons. Notably, our analysis revealed considerable heterogeneity in the electrophysiological properties of neurons across batches of sample preparations and among individual neurons on the same grid (Fig. 3f). To cluster neurons with similar response profiles, we applied Hierarchical Clustering Analysis (HCA) using three parameters: peak value, decay parameter, and reproducibility (Fig. 3g). This approach allowed us to classify neurons into three distinct response

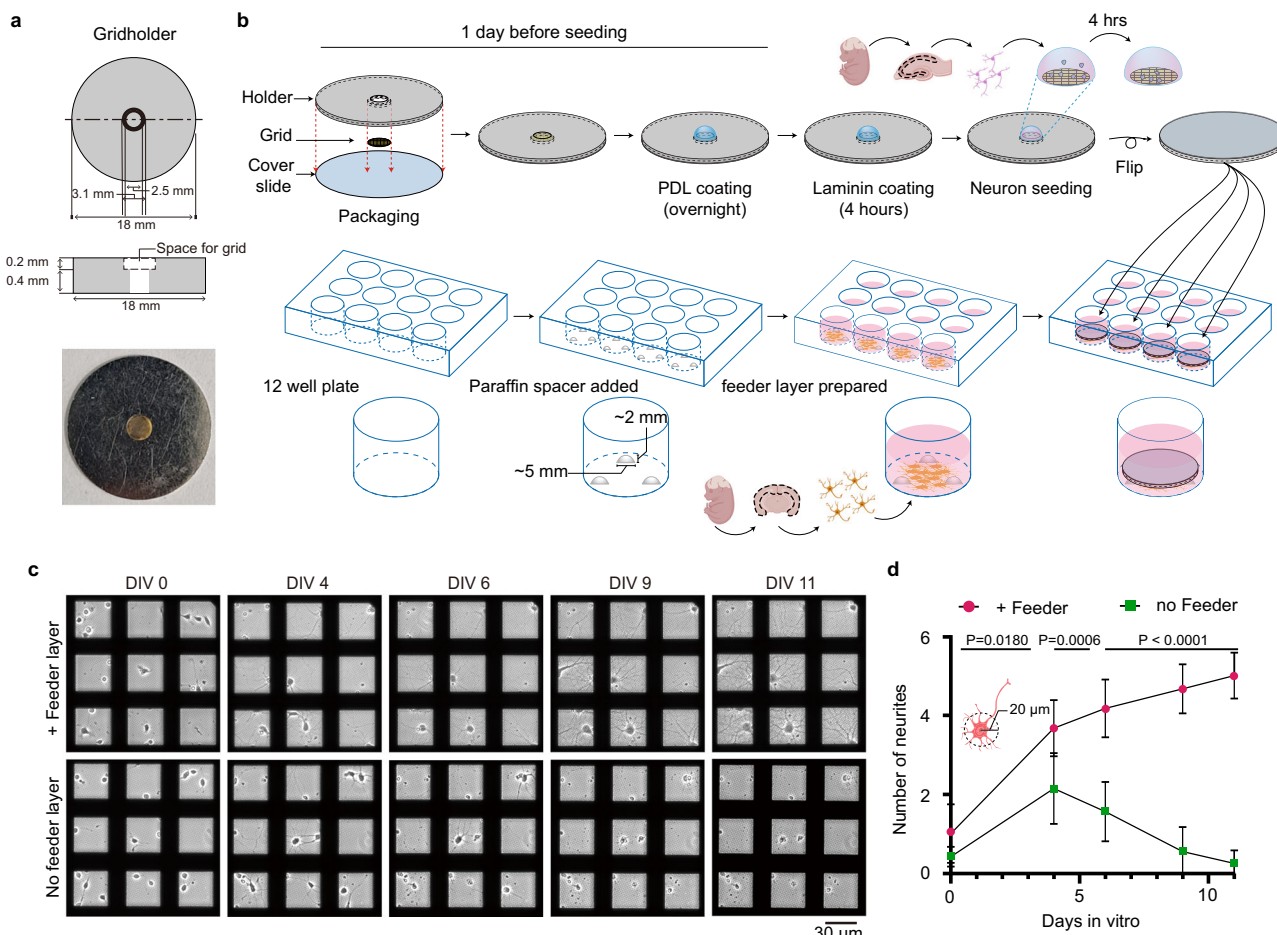

**Fig. 2 | Optimization of neuronal culture for voltage imaging. a** Schematic diagram and photograph of the grid holder for CoVET. **b** A grid was packaged on the central stepped hole of the grid holder and secured with an 18 mm coverslip. 12-well plates were prepared with spacers and astrocyte feeder layers. Four hours after neuron seeding, the packaged grids were flipped and placed in a 12-well plate so that the seeded neurons could face the astrocyte feeder layer. Created in BioRender. Jung, M. (2025) https://BioRender.com/dseff59 **c** Representative phase-contrast images of on-grid cultured neurons with and without the astrocyte feeder layer according to days in vitro (DIV). **d** Number of neurites in 20 μm from each neuron soma was counted according to DIV in both groups cultured with a feeder layer ($n = 10$ separate grids) and without feeder layer ($n = 10$ separate grids). Data were shown as mean ± S.D. Two-sided unpaired t-tests were performed to test statistical significance. Source data are provided as a Source Data file.

clusters: strongly responsive, moderately responsive, and non-responsive (Fig. 3g–i). These clusters exhibited significant differences in the three parameters, highlighting their variability (Fig. 3i). To further validate our experimental design in the context of local electric field variations along the Z axis (Supplementary Fig. 1a, b), we evaluated neuronal responses and clustering patterns based on their somatic coordinates. This analysis revealed no significant differences in response characteristics between neurons located in the central versus peripheral regions, nor among distinct neuronal clusters (Supplementary Fig. 3a–e), which supported the validity of our experimental scheme. These observations suggest that dendritic spatial integration may buffer local differences in field strength, leading to relatively uniform functional output across the network.

By assigning a specific electrophysiological identity to each neuron, we developed a targeted approach for the subsequent cryo-ET experiments. This single-cell approach, centered on electrophysiological properties, minimizes subjective bias and ensures a clearer and more accurate interpretation of electrophysiological responses.

### Correlation of voltage imaging to cryo-ET

After voltage imaging, the entire perfusion chamber containing the grid holder was transferred to a plunge freezer to ensure stable handling of the grids. The grids were then separated from the holders and vitrified using one-sided blotting to minimize physical disturbances to the cultured neurons. The entire process, from voltage imaging to vitrification, was completed within 15 min.

To correlate the results of voltage imaging with the frozen hydrated neurons, the vitrified grids were transferred to a cryo-FIB/SEM system. A BeRST1 fluorescence montage on the grid, incorporating electrophysiological information mapping, was correlated with frozen hydrated neurons through whole-grid alignment with cryo-SEM montages (Fig. 4a). From these correlated montages, we efficiently transferred the electrophysiological information and coordinates of each neuron to the cryo-SEM montages for subsequent analyses. In this process, we could identify 92.6% of neurons whose electrophysiological characteristics were measured. On average, ~7% of neurons from four different grids were lost during transfer and vitrification handling.

To evaluate the multimodal correlation with conventional cryo-CLEM, we used nuclei-specific fluorescence markers and tested them using an integrated Fluorescence Microscope (iFLM) within the cryo-FIB/SEM system (Fig. 4b). This approach allowed us to visualize the nuclei of the neurons identified in both the widefield and cryo-SEM montages. This dual layer of correlation enhances the ability to pinpoint areas of interest within specific neurons based on electrophysiological guidance. To conduct further structural analyses, we milled the soma

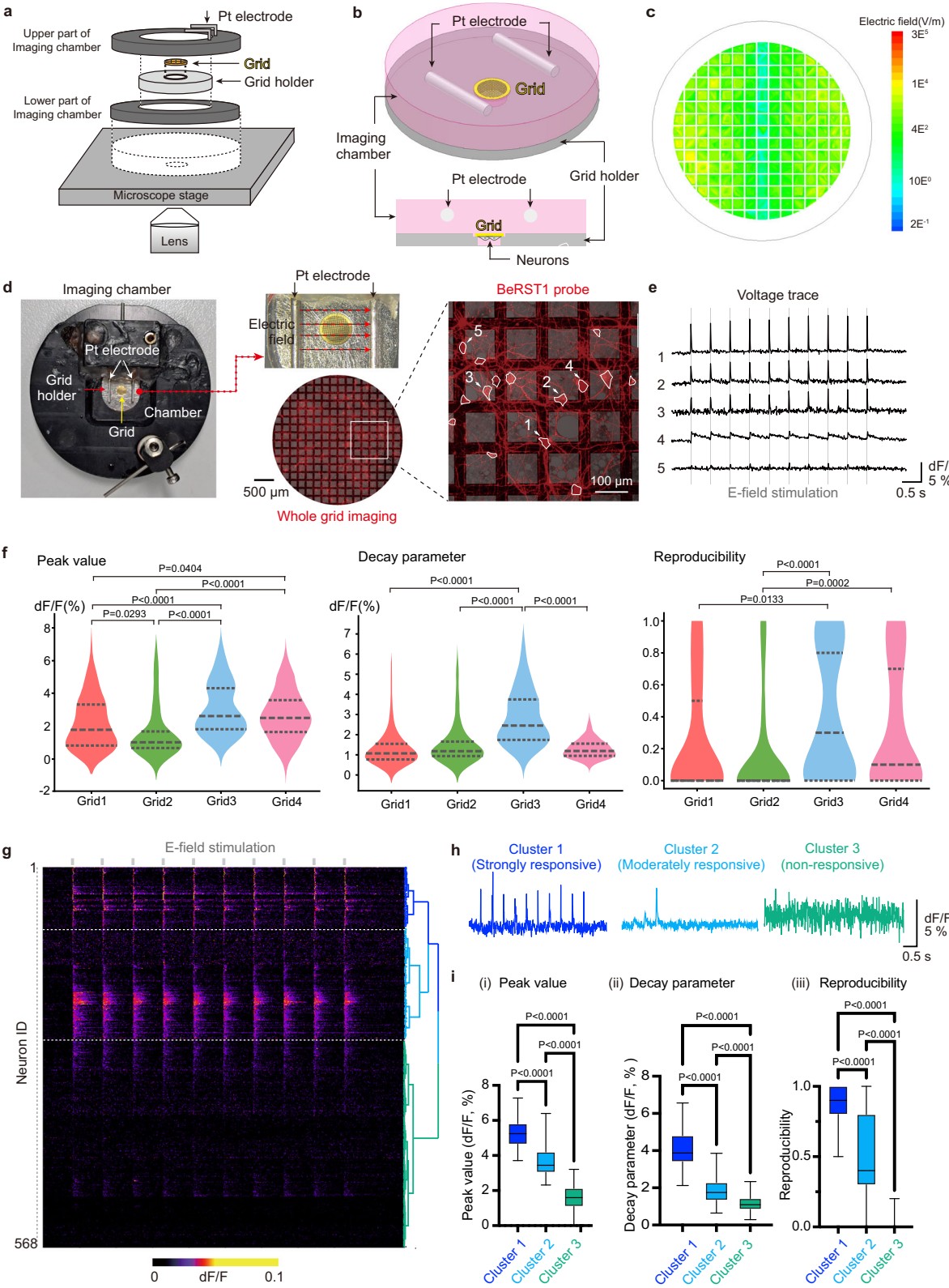

from each cluster using the cryo-FIB/SEM system to facilitate further cryo-ET experiments. The corresponding cryo-ET was performed on each cluster, and tomograms were reconstructed (Fig. 4c). In addition, the individual voltage traces matched the tomograms (Fig. 4c).

### Translational landscapes of ribosomes in different clusters

We demonstrated our combined pipeline by exploring the translational landscapes of ribosomes in relation to neuronal responsiveness.

This approach was informed by previous studies using bulk-level ribosome profiling or proteomics[6,7], which suggested that changes in ribosomal activity are dependent on the electrophysiological activity of neurons[6,7].

We used a combined approach involving voltage imaging and nuclei-specific fluorescent markers to identify the area of interest within the neurons. Subsequently, the cytoplasmic region of each neuronal cluster was selectively milled by cryo-FIB/SEM. All accessible

**Fig. 3 | Measurements of neuronal responses and clustering neurons based on their responses. a** Schematic diagram of voltage imaging configuration. **b** Enlarged view of grid holders integrated with electric field stimulator and imaging chamber. **c** Simulation-based measurement of the applied electric field. **d** On-grid voltage imaging with electric field stimulation. Neurons on a grid were stained with a voltage-indicating dye, BeRST1 (red) (middle), and then placed between platinum wire electrodes to apply an electric field stimulus (left). Changes in the fluorescence intensity were measured for each neuronal soma (right). These experiments were performed on individual grids ($n = 4$). **e** Representative ΔF/F changes according to time from annotated neurons in (**d**). The shaded line indicates the electric field stimulation time. **f** Batch-to-batch variability in the response parameters is plotted as a violin plot. Dashed lines indicated quartiles (Grid1: $n = 106$ neurons; Grid2: $n = 117$ neurons; Grid3: $n = 175$ neurons; Grid4: $n = 169$ neurons). **g–i** Hierarchical clustering analysis (HCA) methods were used to classify neurons' response to 10

electric field stimulation ($n = 568$ neurons from 4 grids). **g** Three clusters were identified using HCA, and are shown as heat maps with dendrograms. **h** Three representative neurons, strongly responsive (blue), moderately responsive (cyan), and non-responsive (green), were represented along with their respective voltage traces. **i** Parameters used for HCA in each cluster were shown as box-and-whisker plots (Cluster1: $n = 90$ neurons; Cluster2: $n = 167$ neurons; Cluster3: $n = 311$ neurons; bold line = median, box = 25–75th percentile, whiskers = min to max). (i) Peak values, (ii) decay parameter, and (iii) reproducibility of action potentials from neuronal responses to electric field stimulation were measured in each cluster and plotted as bar plots. Voltage imaging with electric field stimulation was performed only once per imaging area to preserve neuronal integrity. One-way analysis of variance (ANOVA) followed by Tukey's post hoc test for multiple comparisons was performed to assess statistical significance. Source data for (**f**, **i**) are provided as a Source Data file.

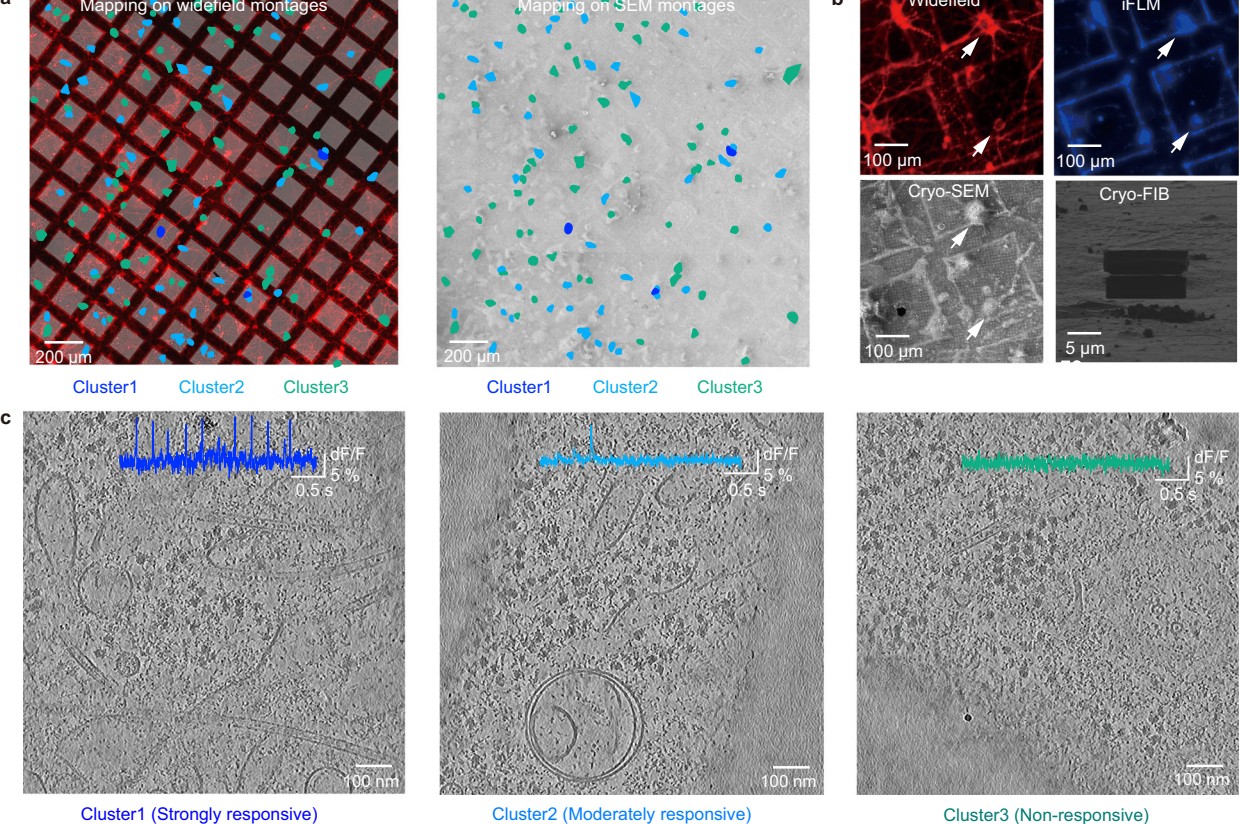

**Fig. 4 | Mapping and imaging clustered neurons on grids. a** Procedure for targeting electrophysiological clusters using cryo-FIB/SEM. Each cluster was mapped onto widefield montages (left), and the coordinates of each cluster in the widefield montages were transferred onto cryo-SEM montages (right). Blue, cluster1; Cyan, cluster2; Green, cluster3. **b** Identification of the nucleus inside the soma using

Hoechst dye, which labels the nucleus. The white arrow indicates the neuronal soma, and the corresponding nucleus is identified in the iFLM image. **a**, **b** were performed on individual grids ($n = 4$). **c** Representative tomographic slices of each electrophysiological cluster. The corresponding voltage traces for each cluster are shown in the tomographic slices (Total 193 tomograms were analyzed).

cytoplasmic areas in the lamellae were imaged using cryo-ET to minimize imaging area bias in the translational landscape analysis. Tomograms of three clusters were reconstructed, and a total of 31,389 particles of ribosome were identified across all tomograms, resulting in the acquisition of a consensus map with 7.8 Å resolution (Fig. 5a, Supplementary Figs. 4, 5).

To gain a detailed structural characterization of the translational landscapes, we performed 3D classification of all ribosomal particles, with a particular focus on key ribosomal binding sites, including the A (aminoacyl), P (peptidyl), and E (exit) sites, as well as the binding sites for elongation factors such as eEF1A and eEF2 (Fig. 5a). This comprehensive analysis revealed eight distinct structural classes of ribosomes, differentiated by their density at the ribosomal binding sites. In

comparison with previously reported structures[32–35], we found that seven of these conformations were aligned with known stages in the elongation cycle: three distinct decoding states (1, 2, and z) corresponding to the codon recognition and sampling steps, two pretranslocation states, and two rotated states associated with the peptidyl transfer step. Additionally, one conformation was identified as representing a translationally idle or inactive state, consistent with a hibernating ribosome containing eEF2 (Fig. 5b, Supplementary Fig. 4).

This conformational information of ribosomes was mapped onto segmented tomograms, enabling the visualization of the in situ translational landscape within each cluster (Fig. 5c). As the mapped tomograms showed distinctive patterns across the clusters, we statistically compared the distribution of the translational landscapes

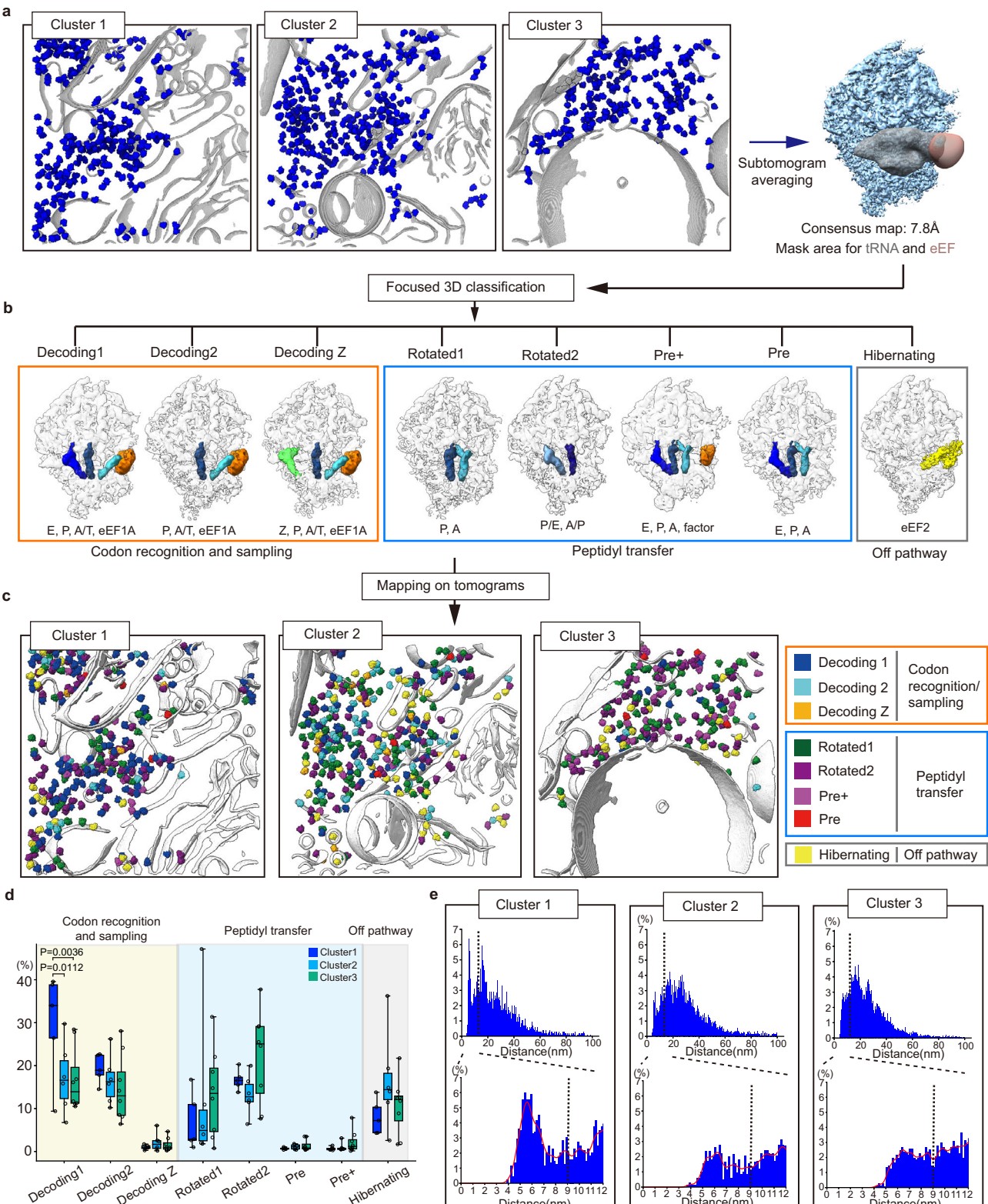

**Fig. 5 | Visualization and statistical analysis of translational landscapes from each cluster. a** Overall scheme of subtomogram averaging. Whole ribosomes from neurons, including 3 clusters (Cluster1 (Strongly responsive), Cluster2 (Moderately responsive), Cluster3 (non-responsive)), were used for subtomogram averaging. **b** Eight different states of ribosomes were observed. **c** Different conformations of ribosomes were mapped in segmented tomograms of each electrophysiologically clustered neuron. **d** Portion of conformational states from each cluster was represented as box and whisker plots with individual data points (bold line: median, box: 25–75 percentile, whiskers: min to max). One-way ANOVA followed by

Tukey's post hoc test for multiple comparisons was performed to assess statistical significance, using the portion of individual neurons as the unit of analysis (Cluster 1: $n = 5$ neurons; Cluster 2: $n = 6$ neurons; Cluster 3: $n = 8$ neurons). For statistical testing only, each data point was weighted by the total number of ribosomes in the corresponding neuron. Source data are provided as a Source Data file.
**e** Histograms of distances between whole ribosomes are shown and dashed lines indicate 12 nm (upper). Distances under 12 nm are shown (lower), with 9 nm, the criteria for polysome[32], indicated by a dashed line. Smoothed plots are shown as red lines.

(Fig. 5d). Notably, the strongly responsive cluster exhibited a significantly higher proportion of ribosomes in the decoding1 state compared to the moderately and non-responsive clusters, whereas the distributions of other states were not statistically different among the clusters (Fig. 5d). To further explore this translational characteristic of each cluster, we analyzed the polysomal states of ribosomes[32,33] by measuring the proximity between ribosomes, specifically the distance between the mRNA exit and entry sites of adjacent ribosomes[33] (Fig. 5e). Interestingly, the strongly responsive cluster displayed a distinct pattern with a sharp peak below 9 nm, which is characteristic of polysomes[32]. In addition, the translational landscapes of the polysomes were statistically analyzed and revealed consistent results in the overall translational landscapes (Supplementary Fig. 6). Collectively, our single-cell electrophysiology-guided in situ ribosomal analysis suggests that translational landscapes and contextual properties differ depending on neuronal electrophysiological responsiveness. Although the direct relationship between responsiveness and translational activity remains to be fully elucidated, our study demonstrates the potential of CoVET to uncover distinctive molecular networks among neurons with different responsiveness.

## Discussion

In this study, we present a pipeline that combines optical electrophysiology with cryo-ET to enable the analysis of molecular structures based on the electrophysiological properties of individual neurons. By utilizing voltage imaging[20–22], we rapidly assessed the electrophysiological properties of all neurons on a grid within minutes while preserving their viability prior to vitrification. Neurons were grouped into distinct clusters based on their electrophysiological profiles. This detailed information, including electrophysiological profiles, cluster identities, and spatial coordinates, was then relayed to the cryo-FIB/SEM system, enabling the selective milling of neurons while maintaining compatibility with conventional cryo-CLEM[17,18]. Subsequent cryo-ET imaging and structural analysis revealed distinct molecular structural features across different electrophysiological clusters.

In our pipeline, we employed a widefield imaging system with the highly sensitive chemical voltage indicator BeRST1[29]. Unlike the viral transduction of genetically-encoded voltage indicators[36], bulk loading of BeRST1 provided near-uniform labeling across all neurons on the grids[29]. Furthermore, we designed a customized grid holder that provided broad compatibility with off-the-shelf live-cell chambers. By integrating a chemical voltage indicator with broadly compatible live-cell chambers, our approach simplifies workflow and enables the measurement of responsiveness in all neurons without the need for genetic modification, making CoVET widely accessible for cryo-ET applications. Integrating CoVET with conventional cryo-CLEM[17] enables the precise identification of regions of interest by combining fluorescence-based protein localization with the characterization of individual neuronal responses. This adds multiple layers of information, enabling precise identification of the regions of interest within the target neurons.

For future application of CoVET, there are several points to be considered. Regarding the electric field stimulation, in our current configuration, lateral platinum electrodes were used to apply electric field stimulation across the neuronal grid (Fig. 3a, b). Simulation and vector decomposition analyses revealed that this setup generates a non-uniform electric field, particularly along the out-of-plane (z) axis (Supplementary Fig. 1a, b). Although we did not observe correlated neuronal responses under our stimulation protocol (Supplementary Fig. 3a−e), such field variability could become more influential under different experimental conditions, such as higher-frequency stimulation, altered culture densities, or varying neuronal maturation states. Systematic validation of stimulation geometries under diverse experimental contexts will be important for optimizing CoVET and ensuring robust interpretation of functional phenotypes.

Furthermore, the choice of clustering parameters is not fixed but can be tailored to the experimental context. While we used peak amplitude in the present study, alternative metrics such as area under the curve (AUC) may offer complementary insights, particularly under different stimulation protocols or imaging regimes.

A variety of factors, including morphological characteristics, neuronal maturation, and networks, are thought to contribute to electrophysiological heterogeneity[37,38]. Prior studies have demonstrated strong correlations between electrophysiological properties and detailed morphological features, showing that the integration of these parameters enables the reliable classification of neuronal subtypes with distinct transcriptomic profiles[39,40]. Moreover, neuronal maturity and synaptic connectivity among the neurons have been shown to strongly influence electrophysiological properties[37,41]. Variations in maturity and connectivity across the population further broaden the distribution of neuronal responses, resulting from both intrinsic excitability and synaptic input from neighboring neurons. In the present study, we observed substantial variability in neuronal responses, both across individual cells and between different culture batches (Fig. 3f). While CoVET enables the correlation of electrophysiological properties with in situ molecular architecture, the absence of complementary biological information at single cell level, such as detailed morphological features, neuronal maturity and networks, may limit the ability to accurately interpret these correlations. Without such contextual information, the heterogeneity among neurons may obscure underlying biological realities, making it difficult to dissect functional relationships at the molecular level. Since we employed a rapid whole-grid screening strategy at low magnification to preserve cellular viability in this study, our imaging inherently lacked the spatial resolution required to resolve fine morphological features such as axonal and dendritic structures. To fully integrate CoVET analysis with detailed morphological characterization and measurements of maturity without prolonged imaging before cryofixation, the development of post-hoc fluorescence analysis compatible with cryo-ET is necessary. Incorporation of post-hoc analysis with CoVET would enable retrospective morphological characterization and measurements of maturity, providing a more comprehensive framework for correlating electrophysiological properties with in situ molecular structures. In relation to neuronal networks, future studies may benefit from combining CoVET analysis with synaptic blockers to isolate cell-autonomous activity, thereby enabling more accurate assessment of intrinsic neuronal responses independent of neuronal networks. Such isolation could also help disentangle intrinsic molecular features from network-driven variability, providing more precise insights into single-cell properties.

In terms of molecular phenotypes, while the expression and localization of ion channels are closely related to heterogeneous neuronal responses[42,43], current technical limitations in cryo-ET present significant challenges for visualizing small membrane-bound complexes, such as voltage-gated sodium and potassium channels (~200 kDa) in situ[44]. These channels are small, embedded in membranes, and are often sparsely distributed, making them difficult to detect in a crowded intracellular environment. Furthermore, comprehensive structural analysis across neurons, including dendritic compartments, is hindered by the limited material volume in dendrites, which makes it difficult to prepare stable lamellae. The cryo-FIB milling of such thin structures requires low throughput and is technically demanding. Considering these constraints, we focused on the ribosomal structures in the soma, which are larger and more abundant complexes amenable to current cryo-ET techniques. Although ribosomes are not directly responsible for electrical activity, their role in activity-dependent translation provides an informative molecular readout of neuronal function[6]. This focus enabled us to capitalize on the strengths of cryo-ET while probing molecular variations across different electrophysiological states. As imaging technologies

advance, enabling the reliable visualization of smaller complexes and finer structures, our CoVET pipeline will be well-positioned to elucidate the structural basis of neuronal responsiveness with high molecular precision.

Collectively, our demonstration of the direct link between electrophysiological heterogeneity and distinct molecular phenotypes highlights the importance of this approach. This strategy can be readily extended to other biological systems and combined with other emerging technologies. Moreover, overcoming current limitations in cryo-ET throughput through advances in sample preparation and data processing, such as high-throughput lamellae fabrication and AI based data processing, will enable CoVET to more precisely resolve the link between molecular and electrophysiological phenotypes. We believe that CoVET, in combination with these technical developments, will play a pivotal role in uncovering diverse molecular mechanisms underlying neuronal responses, including protein translation and synaptic transmission.

## Methods

### Astrocyte feeder layer preparation

All animal procedures were approved by the Seoul National University Institutional Animal Care and Use Committee (IACUC # SNU-230703-1). The astrocyte feeder layer was prepared in a 12-well plate (30012, SPL). For spacing, three flat-bottom paraffin wax pellets (76242, Merck) were placed in each well. They were affixed to their positions by gently pressing the culture plate on an 85 °C heat block for 1 min. Subsequently, each well was coated with 150 μl of 1 mg/ml poly-D-lysine for ~30 minutes at room temperature and then washed twice with distilled water. Primary astrocytes were obtained from the cortices of E18 rat embryos. The cells were suspended at a concentration of 200,000 cells/mL in astrocyte growth medium (NbAstro, BrainBits). A 10 mL cell suspension was loaded into a 75 cm$^2$ cell culture flask (430641U, Corning) and maintained at 37 °C with 5% CO$_2$. The medium was changed every three days. When astrocytes reach full confluence, they were washed twice with calcium-free DPBS (D8537, Merck) and then incubated with trypsin solution (T4174, Merck) at 37 °C for 5 min for harvesting. Harvested astrocytes were either subcultured or frozen in a freezing medium (D9249, Merck) for later use. Two days before primary neuron culture, 1,000,000 astrocytes were thawed and transferred to a 12-well cell culture plate with spacers.

### Cell culture

The HEK293T cells were kind gift from Dr. Jongkyeong Chung (Seoul National University). HEK293T cells were cultured in Dulbecco's modified Eagle medium (Gibco) supplemented with 10% fetal bovine serum(Gibco) and 1% penicillin/streptomycin(Gibco) in a humidified incubator with 5% CO$_2$. Cells with passage numbers below 20 were used for voltage imaging.

### Grid preparation

Quantifoil R2/2 200 mesh, Au (Quantioil) and lacey carbon, 200 mesh, Au grids (Electron Microscopy Sciences) were glow-discharged for 30 s and sterilized using UV light. After sterilization, the grids were packaged using a heat-sterilized customized grid holder and an 18 mm round glass coverslip. The grid was placed into a grid holder such that the carbon film of the grid faced a 2.5 mm window of the grid holder. To attach the coverslip to the back of the grid holder, vacuum grease was applied in three spots using a 1.6 mm paintbrush between the grid holder and the coverslip. Forty microliters (40 μL) of a 1 mg/mL poly-D-lysine solution (P6407, Merck) was applied to the surface of the packaged grid as a drop. The coatings were then incubated overnight at room temperature. Grids were washed with distilled water two times and coated with 40 μL of a 20 μg/mL laminin solution (23017015, Thermo Fisher Scientific) for 4 hours. The solution was not completely removed at every washing step to prevent damage to the grid. The

laminin solution was removed immediately before neurons were seeded.

For HEK293T cells, Quantifoil R2/2 200 mesh Au grids (Quantifoil) were glow-discharged for 30 s and sterilized by UV irradiation. After sterilization, the grids were coated with 50 μg/ml fibronectin, washed three times with PBS, and placed in 35 mm imaging dishes (MatTek). Approximately 150,000 HEK293T cells were seeded onto the grids.

### On-grid neuron culture

Embryonic day-18 (E18) Sprague–Dawley rat embryos were harvested from a pregnant female under isoflurane anesthesia (4% for induction, 2% for maintenance), following the protocol described in previous literature (refs. 7,8). The hippocampi were carefully dissected from the embryos and incubated in 2 mg/mL papain solution (Papain, BrainBits) at 37 °C for 10 min. After enzymatic digestion, the hippocampi were transferred to a hibernating buffer (Hibernate E, BrainBits) and dissociated mechanically using a fire-polished glass pipette. Dissociated cells were centrifuged and resuspended in culture medium (NbActiv1, BrainBits) at a concentration of 100,000 cells/mL. The cell suspension of 33 μL was seeded onto the central hole of the grid packaged holder and incubated at 37 °C in a humidified incubator with 5% CO$_2$ for 3–4 h. The grid with the attached neurons was flipped upside down and placed onto paraffin spacers positioned proximal to the astrocyte feeder layer in a 12-well plate. The neurons were maintained in a growth medium composed of neurobasal medium supplemented with 1% penicillin-streptomycin (P4333, Merck) and 1% N-2 supplement (17502048, ThermoFisher Scientific). Ara-C (C3350000, Merck) at 2 μM was added to the medium at 2–3 days post-plating. One-third of the growth medium was replaced with fresh medium twice a week.

### Evaluation of dendritic development

Phase-contrast images of on-grid neuron cultures with or without a feeder layer were intermittently captured on DIV 0, 4, 6, 9, and 11 throughout the culture period (Eclipse TS2, Nikon). To evaluate dendritic development, the number of dendritic intersections was counted at a distance of 20 μm from the soma within the central 5 × 5 grid frame. After calculating the average number of dendritic intersections per grid for each DIV, the mean and standard deviation were determined for both feeder and non-feeder groups. The analysis was performed using ten grids from each group. An unpaired t-test was performed to compare the two groups at each DIV.

### Electric field simulation

Electric field simulation was conducted using a 3D finite element solver (HFSS, Ansys). All dimensions of the 3D structure were designed to replicate the experimental environment accurately. The 3D model represents the electrical stimulation of neurons, with the grid material modeled as solid Au at a thickness of 20 μm. The mesh consists of square holes measuring 150 μm × 150 μm, with each mesh wire having a width of 20 μm. The area surrounding the grid is filled with the buffer solution. A stimulation signal of 100 V, corresponding to the fundamental frequency of the excitation condition, is applied to a platinum electrode positioned 1 mm above the grid. Electric field analysis is performed at a height of 1 μm above the bottom plane of the grid structure.

### Acquisition of voltage imaging data

Hippocampal neurons were imaged at 10–14 days in vitro. To load fluorescent indicators, the neurons were incubated with 1 μM BeRST1 and 1 μg/mL Hoechst 33258 (Thermofisher) dissolved in an imaging buffer containing 140 mM NaCl, 10 mM HEPES, 30 mM glucose, 3 mM KCl, 1 mM MgCl$_2$, and 2 mM CaCl$_2$ with the pH adjusted to 7.3 with NaOH. After 15 min of incubation, the grid was washed with a fresh imaging buffer, and the coverslip attached to the back of the grid

holder was removed. After attaching a coverslip to the front of the grid holder, it was mounted onto a perfusion chamber with an electrode (CMB-18-EC-PB, Live Cell Instrument) for voltage imaging.

Imaging was performed using an inverted wide-field microscope (Eclipse Ti2, Nikon) with an sCMOS camera (ORCA-Flash 4, Hamamatsu) and a 10× objective lens (numerical aperture: 0.45; CFI Plan Apo Lambda D 10X, Nikon). A high-power red LED (Solis 623 C, Thorlabs) was used as the excitation source for BeRST1, with the irradiance set to less than 18 mW/mm² to avoid photodamage to neurons. The emission of BeRST1 was acquired through a filter cube composed of a 660 nm long-pass dichroic mirror (T660lpxr, Chroma Technology) and a 665 nm long-pass emission filter (ET665lp, Chroma Technology) using an sCMOS camera at a frame rate of 100 Hz.

For electric field stimulation, the perfusion chamber was connected to a stimulus isolation unit (SIU-102B, Warner Instruments) and a multichannel pulse stimulator (Master-9, AMPI) to generate a sequence of trigger pulses (duration: 1 ms, interpulse interval: 500 ms, repeated 10 times). The output current of the stimulus isolation unit was set at 100 mA. For each grid, we imaged 4 chosen field-of-views of $1331 \times 975 \mu m^2$ (2048 by 1500 pixels) for whole grid screening.

### Analysis of voltage imaging data

Intensity traces for individual neurons were extracted using NIS Element software with the binary mask of the neuronal soma manually segmented using VAST software[45]. Using a custom Python script, the intensity traces were converted into dF/F by dividing the trace by the baseline fluorescence level. Using the dF/F dataset from four grids, we quantified the response peak value, decay parameter, and reproducibility of action potentials over a time window of 20 ms before and 480 ms after electric field stimulation (−2 to +48 frames). The response peak value was defined as the maximum dF/F in the initial four frames. To assess reproducibility, we defined it as the proportion of evoked action potentials out of ten repeated stimulations. For this analysis, action potentials were defined as dF/F peak values exceeding 3.5% and a peak signal-to-noise ratio greater than 8 dB. The decay parameter was calculated by subtracting the average of the 2th to 6th frames from the peak value, reflecting the extent of decay as a scalar value. Using the mean values of these parameters in 10 windows as input for hierarchical clustering analysis, the neurons were electrophysiologically clustered into three subgroups: strongly responsive, moderately responsive, and non-responsive.

To identify the effect of an variability in electric field strength on neurons' activity pattern, we defined neurons with x coordinate bigger than −87 μm and smaller than +87 μm as center neurons. Rest of the neurons were defined as peripheral neurons (x-axis was orthogonal to the electrodes). Peak height, decay parameter and reproducibility were compared between two groups using unpaired t-test.

### Vitrification of grids and lamella preparation using cryo-FIB/SEM

After voltage imaging, the grids were immediately vitrified. The entire imaging chamber was used for stable transfer to EMGP2 (Leica Microsystems). One-sided blotting was performed for all vitrification steps to prevent direct contact during the blotting process. The grids were blotted for 5 s. All cryo-FIB milling processes were performed using an Aquillos2 cryo-FIB/SEM instrument and a 35° tilted stage shuttle (Thermo Fisher Scientific). For accurate alignment with fluorescence montages, 6 × 6 cryo-SEM montages were acquired using Maps 3.20 software (Thermo Fisher Scientific). Fluorescence montages with cluster indication were imported into Maps 3.20 and aligned with cryo-SEM montages. Using a 385 nm filter of iFLM (Thermo Fisher Scientific), Hoechst-stained nuclei were imaged. Before cryo-FIB milling, platinum was deposited on neurons for 15 s (1 kV, 10 mA) using a sputter coater, and an organic platinum layer was deposited on the neurons for 20 s using a Gas Injection System (GIS). The cytoplasmic

areas of the neurons from each cluster were milled and thinned to a target thickness of 150 nm with a decreasing current (0.3 nA, 0.1 nA, 50 pA, 30 pA). Five neurons were milled in the strongly responsive cluster, six in the moderately responsive cluster, and eight in the non-responsive cluster.

### Cryo-ET data acquisition

Each lamella was imaged using a 300 kV Krios G4 (Thermo Fisher Scientific) with a Falcon 4i (Thermo Fisher Scientific) and a 10 eV slit SelectrisX energy filter (Thermo Fisher Scientific). The grids were loaded perpendicular to the milling direction. The tilt series was acquired at 2.42 Å/pixel, and the total exposure was 120–140 e⁻/Å² at a defocus of 3–4.5 μm. A dose-symmetric scheme was used for the entire data collection, and considering the pre-tilt angle ±10°, a range of ±40° was acquired in 2° increments. Detailed cryo-ET imaging conditions are summarized in Supplementary Table 1.

### Cryo-ET image processing

All the acquired tilt series were preprocessed using Warp[46], including motion correction and CTF estimation. Motion-corrected frames were used for tilt-series alignment in AreTomo[47]. Aligned tilt series were imported into Warp, and a 4x binned sequential tomogram (9.68 Å) reconstruction was performed. Ribosomes manually picked from 15 tomograms using the Napari Boxmanager plugin were trained using Cryolo 1.9.2. Using the trained model, whole ribosomes from the tomograms of three different clusters were selected. 4x binned subtomogram extraction (9.68 Å) was performed using Warp, and bad particles were excluded using 3D classification with alignment in Relion 4.0[48]. After eliminating bad particles, 3D refinement of the ribosomes was performed. Using M[49], 2x binned subtomograms (4.84 Å) were re-extracted. 3D classification without alignment using masks, including E, P, A, and elongation factor binding sites, was performed, and eight different states of the ribosomes were classified. To reconstruct the consensus map, unbinned subtomograms (2.42 Å) were extracted, averaged, and refined using M[49] and Relion 4.0[48]. All structures were visualized using ChimeraX[50], and tomograms were visualized using IMOD[51].

### Spatial analysis and statistical analysis of translational landscapes

Mapping of each ribosome from different translational states was performed using subtomo2chimerax (https://github.com/builab/subtomo2Chimera), and the segmentation of each tomogram was performed using MemBrain[52]. Visualization of ribosomes and segmented tomograms was performed using ChimeraX[50]. Distances between ribosomes were calculated using a polysome analysis script (https://github.com/mvanevic/polysome_mef) by modifying the coordinates of the mRNA exit and entry site[33]. The number of particles from each electrophysiological cluster was calculated using the star files from each class. Statistical analysis of the translational landscapes from each cluster was conducted using a custom Python script that performed a one-way ANOVA followed by Tukey's post hoc test on the proportions of each conformational state from single cells across different clusters, weighted with the total ribosomes in each cell. Weighted mean, standard deviation and effective sample size were used for statistical analysis.

### Reporting summary

Further information on research design is available in the Nature Portfolio Reporting Summary linked to this article.

## Data availability

Source data underlying Figs. 2d, 3f, 3i, 5d and Supplementary Fig. 3 and Fig. 6 are provided in Source Data file. A consensus map of ribosomes and maps of related conformations generated in this study were

deposited in the EMDB under accession codes EMD-61318 (Consensus map), [https://www.ebi.ac.uk/pdbe/entry/emdb/EMD-61318], EMD-61633 (Decoding1 state), [https://www.ebi.ac.uk/pdbe/entry/emdb/EMD-61633], EMD-61634 (Decoding2 state), [https://www.ebi.ac.uk/pdbe/entry/emdb/EMD-61634]. EMD-61635 (DecodingZ state), [https://www.ebi.ac.uk/pdbe/entry/emdb/EMD-61635], EMD-61636 (Hibernating state), [https://www.ebi.ac.uk/pdbe/entry/emdb/EMD-61636], EMD-61637 (Pre-state with factor), [https://www.ebi.ac.uk/pdbe/entry/emdb/EMD-61637], EMD-61638 (Pre-state without factor), [https://www.ebi.ac.uk/pdbe/entry/emdb/EMD-61638], EMD-61639 (Rotated1 state), [https://www.ebi.ac.uk/pdbe/entry/emdb/EMD-61639], EMD-61640 (Rotated2 state), [https://www.ebi.ac.uk/pdbe/entry/emdb/EMD-61640]. Raw tomograms generated in this study were deposited in EMPIAR under accession code EMPIAR-12948. Source data are provided with this paper.

## Code availability

Code used for voltage imaging data analysis is available at [https://github.com/Dangallll/CoVET].

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

## Acknowledgements

This work has been supported by the Korean National Research Foundation (RS-2021-NR056571, RS-2020-NR049538, RS-2020-NF000307, RS-2024-00344154, RS-2024-00440289, RS-2025-00559184) and SUHF Foundation to S.-H.R., Korean National Research Foundation (2020M3C1B8016137, RS-2024-00334680, RS-2024-00436783) to M.C., and Korean National Research Foundation (RS-2024-00411334) to M.J. Cryo-EM data were collected and processed at the Center for Macromolecular and Cell Imaging (CMCI), Institute for Basic Science (IBS), Kaist Analysis for Research Advancement (KARA), and Global Science Experimental Data Hub Center (GSDC) at the Korea Institute of Science. We thank the members of the CMCI and Neurophotonics lab for their discussions and suggestions.

## Author contributions

M.J. conceived and designed the project, performed cryo-FIB/SEM milling, data collection, structural analysis, wrote the manuscript, and contributed to the review. G.K. designed the project, performed grid preparation, voltage imaging, established neuron-glia co-culture, processed voltage imaging data, wrote the manuscript, and participated in the review process. D.L. performed voltage imaging and participated in the review process. S.K. (Institute of molecular biology and genetics, Seoul National University) conceived the project, established the neuron-glia co-culture, and performed voltage imaging. S.K. (Department of Semiconductor, Gachon University) performed electric field simulations under the supervision of Y.K. M.C. supervised the study and wrote the manuscript. S.R. supervised the study and wrote the manuscript.

## Competing interests

The authors declare no competing interests.
