## [Transparent Peer Review file · Nature Communications]

Correlative Voltage Imaging and Cryo-Electron Tomography Bridge Neuronal Activity and Molecular Structure

Corresponding Author: Professor Soungun Roh

Version 0:

Reviewer comments:

Reviewer #1

(Remarks to the Author)

In this manuscript, the authors have developed a pipeline for collecting the electrophysiological properties of neurons using voltage imaging before conducting cryo-ET imaging. This approach has the potential to explore correlations between the functional properties and molecular structures of neurons. To showcase the capability of this technique, the authors investigated the relationship between the composition of various translational profiles of ribosomes and the responsiveness of cultured neurons to electrical stimulation. They found that the strongly responsive cluster of neurons exhibited a significantly higher prevalence of decoding 1 state compared to the moderately and non-responsive neurons.

While the concept of drawing correlations between functions and molecular structures is inherently intriguing, the current demonstration regarding the correlation between electrical responsiveness and translational profiles of ribosomes lacks compelling evidence.

(1) The responsiveness of cultured neurons to electrical stimulation can vary significantly, influenced by many factors, such as cellular morphology and the local extracellular environment (Gorazd Pucihar et al., IEEE Trans. Biomed. Eng. (2009), Vincent Vilette et al., Cell (2019)). Consequently, clustering based solely on responsiveness may not accurately reflect the actual electrophysiological state of the neuron. Furthermore, the electrophysiological state of neurons is more closely linked to the expression patterns of various types of ion channels distributed throughout the cells, including both the cell soma and neurites. Focusing solely on molecular structures within the soma may not provide a comprehensive understanding to elucidate the observed electrophysiology state.

(2) The observed differences in the translational profiles of ribosomes are more likely a consequence of electrical stimulation, rather than a direct correlation with the intrinsic electrophysiological state. As the authors noted, electrical activity can induce translation. It's reasonable to expect that the responsive neurons were more strongly perturbed by the electrical stimulation and consequently triggered changes of the translational profiles in them.

(3) The term "AUC" is commonly associated with ROC curve in classification. I would suggest using a different term to avoid confusion.

(4) I cannot find any description regarding the measure of "reproducibility"

Reviewer #2

(Remarks to the Author)

Summary

This study represents the first example to my knowledge of electrophysical/chemical information being used to study cells in a correlative manner for study using cryo electron tomography. The ability to use this information is novel and represents a significant addition to the field.

The authors further used this information to inform the latent translational profiles of pre-characterised cells, leading to low resolution structures of ribosomes in different stages of translation which the authors could then link to the electrophysical profile of the specific cell. This demonstrates that the technique could be used to unpick structural details using electrophysiology as a selection criterion.

The ribosomes studies, while interesting, are nevertheless at a relatively low resolution compared to other studies on ribosome translational dynamics and the information gained on the structures is limited to sub-unit level insight. This is not a critical consideration for the work, however.

In general, this study is worthy of merit and could be used as a basis to study various other neuronal phenomena where electrophysiology has a direct impact on cellular function. I really enjoyed the work and especially the concept behind it.

Major Comment

The major change to this manuscript should be in the "Preparing primary neuron for effective CoVET analysis" section. The use of co-cultures to study neurons is not novel, including where the cultures are to be used for cryo electron tomography. See 10.1126/science.1261197 for an example. It would therefore be useful for the authors to tone down the novelty of this aspect in the results and focus more specifically on the need to grow cells at low density for imaging and incorporate this into the results for the need to design a new sandwich assembly.

Minor Comments

The sentence "Given the correlation between electrophysiological properties and molecular society of neurons, single-cell analysis techniques based on electrophysiological properties, such as Patch-seq and Voltage-seq^{11,12}, are emerging as essential tools to characterize dynamic molecular changes to the heterogenous electrophysiological property" should be reworded to remove molecular society and remove the reference to electrophysiology being essential tools to characterise molecular details. It does not.

Multiple references/uses of phrase "molecular society" should be re-worded as it is ambiguous in meaning.

There is no need to always capitalised the first letters of acronyms when written out i.e. In situ cryo-Electron Tomography (cryo-ET). There several instances of this.

"samples culture method" the word method is redundant.

"One day prior to seeding neurons, a grid was mounted onto the grid holder" Is this a necessary step?

Figure 1: Might be useful to restructure/alter the figure to make it clearer that the neuronal activity characterisation is done elsewhere, analysed and then the results incorporated into the correlation.

"We demonstrated our combined pipeline by exploring the translational landscapes of ribosomes within electrophysiological context." The use of electrophysiological context is odd and should be reworded.

As I touched on in the summary, the resolution of the ribosome structure generated is lower than similar studies investigating in-situ ribosomes (DOI: 10.1038/s41592-020-01054-7; 10.1038/s41467-023-36372-9; 10.1126/science.adh1411). Some of these studies use fewer particles with equivalent equipment and pixels sizes yet achieve a higher resolution. Can the authors comment on why they were unable to replicate these published results? With ~20000 particles I would expect a better resolution for the global structure.

Can the authors proved an FSC curve?

Will the authors deposit the data on EMPIAR?

"greatly simplifying the workflow and making CoVET universally accessible to the cryo-ET applications" This sentence should be reworded and an explanation of what the simplification step is.

"Moreover, integrating CoVET with conventional cryo-CLEM17 adds multiple layers of information, enabling precise identification of the regions of interest within the target neurons." This sentence should be re-written as it is not clear what the multiple layers of information are, or is the precision of identification of regions using this approach characterised.

"By using CoVET to pinpoint neurons with electrophysiological defects and then performing in situ structural analysis, we can directly correlate functional abnormalities with structural phenotypes, such as amyloid beta or tau aggregation¹⁴." This sentence should be either removed or changed as the claim to be able to use this to target Abeta or Tau is unproven – speculation over applications should be more clearly worded.

"Ribosomal landscapes" What does that mean?

The manuscript should be proof read and grammatical errors corrected.

Reviewer #3

(Remarks to the Author)

Jung et al. developed a technique they call Correlative Voltage Imaging and cryo-ET (CoVET). They combined voltage imaging of cultured neurons with subsequent FIB milling and cryo-electron tomography. The authors designed specialized neuron culture and imaging methods that enable mapping between real-time voltage imaging traces and in situ cryo-ET images. They report significant variations in ribosome states among neurons with differing excitability.

The ability to correlate physiology with ultrastructure is an important goal, and this paper provides a technical advance in that direction. However, the description of the technology is insufficiently detailed, the statistics are weak, and the biological claims are unconvincing. A manuscript focused on the technical aspects of the measurements that avoids making unconvincing biological claims could be suitable for publication.

1. Many of the key details were left for the Methods, but since this is a paper specifically about a new method, it would make sense for all the non-standard aspects to be in the main text. Here is some information that is in the Methods which should be in the main text.

Please state the composition of the TEM grid: what was the surface material, and what were the support bars made of? Was it cleaned or treated prior to addition of PDL and Laminin?

Please specify the seeded cell density (cells/cm²) rather than saying just "one-fifth of the conventional neuronal culture," since different labs have different conventions.

Please explain how the coverslip was attached to the back of the grid and the grid holder.

How were the paraffin dots produced and applied to the bottom coverslip?

What were the nuclei-specific fluorescence markers? When and how were these delivered?

It would be helpful to have a figure describing the cryoET workflow too, since for non-experts this seems quite complex.

2. Statistics:

Fig. 1d) How many biological replicates, i.e. independent plating rounds, was this? There can be substantial batch-to-batch variability in neuron culture survival, so this needs to be tested across replicate experiments.

Fig. S2 is crucial context on the batch-to-batch variability and should be in the main text.

The clustering into 3 classes is not sufficiently justified. Does the distribution of ephys properties actually fall into three distinct groups, or is it a broad distribution which is being divided up? The authors should plot scatter plots of the neurons for all pairwise combinations of parameters to determine whether there are really 3 clusters.

p. 9: in the statistics of neurons found and neurons lost during vitrification, how many independent grids was this procedure performed on?

Please give statistics on the numbers of neurons subjected to cryoET in each cluster, and the distribution of numbers of ribosomes per neuron. How many biological replicates (independent platings and measurements) were the data recorded from? Ah, the information is buried in the supplement: The whole paper is based on measurements on groups of 4, 3, and 5 neurons from the three clusters, respectively (presumably from a single grid?). In this case, all of the bar graphs should indicate the specific individual values for each neuron. This is unlikely to be enough cells to make any firm biological conclusions about the relation between the ephys properties and the ribosome properties.

For the mapping of ribosome conformations onto ephys states, were the p-values corrected for multiple-hypothesis testing? They should be. Was the n value used the number of ribosomes or the number of cells?

3. Electric field stimulation: I'm very confused about the electric field geometry. It looks like the stim electrodes are right above the stainless steel disk, in which case the disk will shunt most of the electric fields around the neurons. Furthermore, if the neurons are plated on a conducting grid, the grid would further shunt most of the electric field. The authors must provide a clearer view (e.g. a cross-section of the geometry during stimulation) and should provide 3-D finite-element simulations to model the electric field.

4. Biology: The authors do not sufficiently explain on the significance of the observed three groups of neurons. Rat hippocampal cultures are composed of heterogeneous neuron types, each exhibiting distinct electrical behaviors. Can authors identify the major factors behind these groups? Are they associated with differences in neuron types, maturation states, morphology, or health conditions? The absence of such information weakens the interpretation of the subsequent characterization of ribosome states. Here are also suggestions on experiment details:

1) Differences in spike height are unexpected, as triggered action potentials should be consistent. The authors should rule out the possibility that this variability reflects the role of background autofluorescence..

2) Peak height, Area Under the Curve (AUC), and reproducibility appear to be strongly correlated, as shown in Supplementary Figure 2. It is not clear if these two parameters give independent information.

3) When characterizing intrinsic excitability, synaptic blockers are needed to block communication between neurons. Imaged neurons could be silenced or excited by their neighbors.

4) The authors characterize the distribution of ribosome states at the neuron cluster level. How accurately can the ribosome patterns of individual cells predict their electrical signatures? How accurately can the electrical signatures of individual cells predict their ribosome patterns? This analysis would strengthen the conclusions on the proposed relationship between ribosome states and electrical behavior.

5) Could the authors provide more functional insights into the observed ribosome states? For instance, if channel proteins are being preferentially translated, would it be possible to observe an enrichment of ribosomes or polysomes near the cell membrane, endoplasmic reticulum (ER), or secretion-related organelles? Such spatial correlation could provide stronger evidence linking ribosome states to specific functional outcomes and enhance the biological relevance of the findings.

Minor

1. In Main text paragraph 1, please provide more discussion on the importance of structure characterization rather than RNA-based characterization.

2. Please Provide the full name of TEM (Transmission electron microscopy).

3. In Analysis of voltage imaging data, should be "8 decibel of peak signal to noise ratio".

4. Overall the English needs improvement (e.g. the authors repeatedly use the word "society" in a way that doesn't make sense).

Reviewer comments:

Reviewer #1

(Remarks to the Author)

The revised manuscript places a stronger emphasis on the technical advancements for establishing correlations between the electrophysiological properties and molecular structures of neurons. Additionally, the authors have improved the presentation and interpretation of the biological observations. I have no additional concerns.

Reviewer #2

(Remarks to the Author)

I am happy with the changes made to the manuscript.

Reviewer #3

(Remarks to the Author)

In this revised manuscript, Jung et al. have reorganized the text to provide a more comprehensive description of the CoVET technique. The authors provide more details in their specialized neuron culture, voltage imaging and in situ cryo-electron tomography (cryo-ET) approaches. The integration of electrophysiology with ultrastructural imaging represents a promising step forward and could serve as a valuable resource for future users and applications.

The authors report variations in ribosome states among neurons with differing excitability; however, the correlations between electrophysiological features and ribosome states are weak. Considering the low throughput of current cryo-ET, it would be valuable for the authors to discuss how this technique can be applied to address biological questions, or how the throughput could be increased.

Major:

QIII-4-1/RIII-4-1:

The width of individual action potential is typically a few ms, whereas the interspike intervals are typically much longer. Given that the duration of electrical field stimulation is only 1 ms, it is unlikely to induce multiple spikes within a 10 ms window.

I suspect that the observed variation in peak amplitude may be influenced by the low frame rate of the imaging system. A 10 ms sampling interval may fully capture narrow spikes but only partially capture wider ones, leading to inconsistent peak measurements. I recommend that the authors consider using the area under the curve (AUC) of each spike as a more robust measure of spike amplitude, especially given the low temporal resolution of their voltage imaging.

QIII-4-3/RIII-4-3:

The low-magnification images (Fig. 4a & b) suggest that neurons are likely interconnected via synapses. While spontaneous synaptic activity may be minimal in low-density cultures, synaptic modulation could become significant when neurons are actively spiking under electric field stimulation. As a result, the absence of synaptic blockers is still concerning. Synaptic inputs from neighboring neurons could disturb the interpretation of intrinsic properties, such as reproductivity, by activity dependent modulation from excited neighboring neurons.

Given the low throughput of cryo-ET and the relatively weak correlation between electrophysiological features and ribosome states, capturing broad phenotypes that are a mixture of intrinsic and extrinsic factors (authors use "functional output state of each neuron within its microenvironment", which is confusing) makes it more difficult to have clear biological conclusions. As a methods-focused manuscript, the authors should emphasize the importance of experimental design and controls for future users applying this technique.

QIII-3:

The geometry of the e-field setup is still confusing. Fig. 1.III suggests that the neurons are on the opposite side of the grid from the field-stim electrodes. Is this true? Fig. 3b is ambiguous about where the cells are. Please make clear whether the cells are on the top or bottom side of the grid in the cross-sectional view. The cross-sectional simulation in Fig. RI-1-i.b shows big electric field on one side of the grid, and much smaller on the other side, as one would expect from grid shielding. Unclear which side of the grid the map in panel c (which is the only thing shown in the manuscript) corresponds to. The authors need to characterize the variability in the electric field strength across the locations where the neurons are measured.

There is big variation in E-field distribution based on the simulation results. The center of grids is around the 10V/m while the edge can achieve 1×10^4 V/m. Does this large difference in electric field dominate the variance observed for neural activity? The authors should provide some spatial maps showing the distribution of key electrophysiological features, including cluster types, peak value, decay parameter, reproducibility and spike numbers as a function of location on the grid.

Also, the simulations lack information about the vectorial aspect of the electric field. Since the neurons are extended in the x-y plane, but comparatively flat in the z direction, in-plane vs. out-of-plane electric fields will have very different effects on the cells. Please separately show the in-plane and out-of-plane components. If, as I suspect, most of the electric field is in the z-direction, then a much more homogeneous electric field could be obtained by putting a single electrode directly above the

sample (or a mesh, to permit light to go through), and applying the pulses between the sample chuck and the topside electrode, i.e. setting up a parallel-plate capacitor arrangement. The near-null in the electric field along the midline of the sample is consistent with my interpretation. An alternative to redesigning the sample chamber is just to connect the two Pt wires together, and to apply the voltage pulse between these two wires and the sample chuck.

Other:

Fig. 3f, middle: Why does "Decay parameter" have units of dF/F ?

Minor:

QIII-1-2 & Line 95 in manuscript: The use of "cells/mL" to describe the seeding density of a 2D primary neuron culture is confusing. (Is that the concentration of resuspended neurons?) Authors also used cells/cm² when demonstrating their low-density culture, which is inconsistent with literature standards.

Abstract and line 87: The authors refer to ribosomes having different "contextual information". Unclear what this means. Please rephrase to be more clear and specific.

Line 635: "Each column is shown with mean \pm s.e.m." It took me a while to figure out that the dashed horizontal lines are supposed to represent these values, rather than individual data points. Consider showing mean and variability with a box. Also, it looks more like s.d. to me than s.e.m. Please check.

Line 397: still refers to AUC instead of "decay parameter".

Line 401: The definition of reproducibility doesn't make sense. Please check this sentence carefully, some words are missing.

Version 2:

Reviewer comments:

Reviewer #3

(Remarks to the Author)

The authors have reasonably addressed the critiques and the manuscript can now be published.

Open Access This Peer Review File is licensed under a Creative Commons Attribution 4.0 International License, which permits use, sharing, adaptation, distribution and reproduction in any medium or format, as long as you give appropriate credit to the original author(s) and the source, provide a link to the Creative Commons license, and indicate if changes were

made.

Point-by-point response to the reviewers' comments (NCOMMS-24-67164-T)

Jung and Ko et al,

We sincerely appreciate the reviewers' constructive comments regarding our manuscript. We acknowledge that our initial interpretation of the relationship between electrophysiological states and translational landscapes may have been overstated, due to the lack of accompanying biochemical profiling and relatively limited statistical power.

To address these concerns, we performed additional biological replicates and rigorous statistical analyses, resulting in stronger statistical support. While these efforts enhanced the robustness of our results, we recognize that further biochemical validation would be necessary to fully substantiate the biological interpretations.

However, as the primary focus of this manuscript is the development and demonstration of the CoVET method, we have refrained from extending into additional biochemical assays in the current study. Instead, we revised the manuscript to present the differences in translational landscapes among neuronal clusters in a more neutral and descriptive manner and emphasize the methodology itself by including methodological details in the main texts.

There is another minor change in the revised manuscript. In the initial manuscript, we included the grids which were not imaged using cryo-ET for robust electrophysiological clustering with sufficient number of neurons. Through additional biological replicates, we could ensure sufficient number of neurons for electrophysiological clustering without the grids which were not imaged using cryo-ET and exclude them for the clustering. Therefore, more relevant and clear correlation could be achieved.

We believe that CoVET offers a powerful framework for linking electrophysiological properties with molecular architecture at the single-cell level, and that it holds strong potential to advance the understanding of neuronal structure–function relationships. We are deeply grateful for the reviewers' careful evaluation, and we address each comment point-by-point as detailed below.

Reviewer #1 (Remarks to the Author):

In this manuscript, the authors have developed a pipeline for collecting the electrophysiological properties of neurons using voltage imaging before conducting cryo-ET imaging. This approach has the potential to explore correlations between the functional properties and molecular structures of neurons. To showcase the capability of this technique, the authors investigated the relationship between the composition of various translational profiles of ribosomes and the responsiveness of cultured neurons to electrical stimulation. They found that the strongly responsive cluster of neurons exhibited a significantly higher prevalence of decoding 1 state compared to the moderately and non-responsive neurons.

While the concept of drawing correlations between functions and molecular structures is inherently intriguing, the current demonstration regarding the correlation between electrical responsiveness and translational profiles of ribosomes lacks compelling evidence.

We thank the Reviewer for being interested in our study and for recognizing its significance.

We have revised the manuscript to address all the points raised in the review, as described below.

QI-1-i. The responsiveness of cultured neurons to electrical stimulation can vary significantly, influenced by many factors, such as cellular morphology and the local extracellular environment (Gorazd Pucihar et al., IEEE Trans. Biomed. Eng. (2009), Vincent Villette et al., Cell (2019)). Consequently, clustering based solely on responsiveness may not accurately reflect the actual electrophysiological state of the neuron.

RI-1-i. Thank you for your insightful questions and we understand and share the reviewer's concerns regarding this issue. The references you mentioned suggest that the transmembrane potential is asymmetric throughout the cell, influenced by the orientation of the applied electric field, morphology and local extracellular environment.

As highlighted by the reviewer, detailed morphological characters including structure of axon and dendrites are strongly correlated with electrophysiological properties and could influence the neuronal responses¹. Furthermore, combining electrophysiological and morphological properties enable detailed neuronal subtype clustering, and each subtype exhibits a distinct gene expression profile^{2,3}. However, in our experimental scheme, low magnification imaging for rapid screening of whole neurons on the grids is necessary to maintain the neuronal viability before cryo-fixation, which is insufficient to characterize the detailed structure of axon and dendrites. Although, by reducing the coverage of neurons on the grid to be imaged, high magnification imaging for morphological characterization is possible, this makes it challenging to confirm sufficient numbers of neurons for structural analysis, which is a crucial aspect in cryo-ET. The most ideal way to characterize morphological features is using post-hoc analysis of individual neurons as the previous studies combining electrophysiology and morphological characteristics². By utilizing post-hoc analysis, the morphological features of whole neurons on the grids could be analyzed^{2,3}. However, the post-hoc analysis after cryo-ET imaging is under development and the influence of platinum and organic platinum coating in cryo-FIB/SEM milling procedure on post-hoc analysis should be validated. Once a robust post-hoc technique is established, combining it with our approach will enable a comprehensive examination of the structural phenotype and neuronal subtypes.

Nevertheless, minimally, to assess whether morphological features and local environment contributed to variability in our dataset, we analyzed the relationship between size of soma and local neuron density with respect to electrophysiological response parameters. The area of each neuron was calculated by counting the number of pixels occupied by each segmented neuron soma and the local density was computed by counting the number of neurons within a radius of 300 pixels, corresponding to 195 μm . All resulting R^2 values were below 0.1 (Fig. R1-1-ia), suggesting negligible correlation under our experimental conditions.

Moreover, in terms of neuronal response only, a previous study employing a stimulation about the responsiveness of primary neurons to electric field stimulation⁴, which used a stimulation scheme identical to ours, demonstrated no significant orientation dependent response in electric field stimulation across the overall population of 2D cultured primary neurons. This outcome is likely due to the integrated effect of each subcompartment in generating the action potential while local asymmetry in membrane potential could be present and could explain

the negligible correlation in our dataset.

To further address this concern, we conducted simulations of the electric field distribution on the grid and confirmed that the neurons are exposed to a field strength of $\sim 1,089$ V/m, which is sufficient to evoke action potentials⁵ (Fig. RI-1-ib, c). These results support that our stimulation paradigm reliably activates neurons under our experimental conditions. We have incorporated these rationale into the revised Results and Discussion section.

“Results

~To validate that our electric field stimulation system with a grid holder can deliver adequate stimulation, we performed electric field simulations (Fig. 3c). Our simulation indicated that, on average, a 1,089 V/m electric field was applied to the neurons on the grids, which is sufficient to evoke action potentials⁵. Furthermore, previous study revealed that isotropic responses across neurons are induced by electric field stimulation in the 2D primary cultured neurons⁴. Based on this information, we validated that our system is capable of evaluating the responses of single neurons on the grids. ~”

“Discussion

~A variety of factors, including morphological characteristics and neuronal maturation, are thought to contribute to electrophysiological heterogeneity^{1,6}. Prior studies have demonstrated strong correlations between electrophysiological properties and detailed morphological features, showing that the integration of these parameters enables the reliable classification of neuronal subtypes with distinct transcriptomic profiles^{2,3}. Moreover, neuronal maturity has been shown to strongly influence electrophysiological properties^{6,7}, and uneven maturation across the population further broadens the distribution of neuronal responses. In the present study, we observed substantial variability in neuronal responses, both across individual cells and between different culture batches (Figure 3f). While CoVET enables the correlation of electrophysiological properties with in situ molecular architecture, the absence of complementary biological information at single cell level, such as detailed morphological features and neuronal maturity, may limit the ability to accurately interpret these correlations. Without such contextual information, the heterogeneity among neurons may obscure underlying biological realities, making it difficult to dissect functional relationships at the molecular level. Since we employed a rapid whole-grid screening strategy at low magnification to preserve cellular viability in this study, our imaging inherently lacked the spatial resolution required to resolve fine morphological features such as axonal and dendritic structures. To fully integrate CoVET analysis with detailed morphological characterization and measurements of maturity without prolonged imaging before cryo-fixation, the development of post-hoc fluorescence analysis compatible with cryo-ET is necessary. Incorporation of post-hoc analysis with CoVET would enable retrospective morphological characterization and measurements of maturity, providing a more comprehensive framework for correlating electrophysiological properties with in situ molecular structures.~”

Fig. RI-1-i. a, Pairwise plot between response parameters and morphological parameters b, Overall applied electric field stimulation. Upper panel indicates the applied electric field within the whole chamber. Lower panel indicates a magnified view of the applied electric field within the grid holder. c, Applied electric field to neurons on grids.

QI-1-ii. Furthermore, the electrophysiological state of neurons is more closely linked to the expression patterns of various types of ion channels distributed throughout the cells, including both the cell soma and neurites. Focusing solely on molecular structures within the soma may not provide a comprehensive understanding to elucidate the observed electrophysiology state.

RI-1-ii. We greatly appreciate the reviewer's insightful comment emphasizing the importance of examining molecular structures distributed throughout the neuron, including both soma and neurites, for a comprehensive understanding of the relationship between structure and electrophysiological states. We fully acknowledge that the electrophysiological properties of neurons are closely linked to the expression and localization of various ion channels across subcellular compartments, beyond the soma alone.

However, current technical limitations in cryo-ET imaging pose significant challenges in visualizing smaller molecular complexes such as sodium or potassium ion channels (~200 kDa). The intrinsic molecular heterogeneity and complexity within cells further complicate the direct visualization of these channels in situ, and no references have yet demonstrated their successful visualization within neurons using cryo-ET. Recent studies try to visualize AMPA receptors (~500 kDa) using Cryo-ET, however, the resolution is not sufficient to interpret accurate biology by the protein structure⁸. Once technical advancements enable high-

resolution visualization of these smaller complexes, our methodology will be ideally positioned to elucidate the correlations between molecular architecture and neuronal responsiveness comprehensively.

Considering these limitations, we have chosen to focus our study on analyzing the translational landscapes of single neurons, leveraging the established link between neuronal activity and protein synthesis^{9,10}. While examining molecular structures throughout the entire neuron, including dendrites, would indeed provide a more informative analysis, cryo-FIB milling of thin neuritic structures currently poses another significant technical challenge. Dendrites typically do not contain sufficient material volume to generate stable lamellae without high-density bundling¹¹, which subsequently compromises our ability to correlate specific dendritic structures with their originating soma and electrophysiological properties at the single-cell level. Future work involving sparse expression of fluorescent proteins will allow us to trace individual dendrites back to their originating somas within bundled neurites. This allows us to extend the correlation between molecular structural states and electrophysiological properties throughout the neuron.

Moreover, as cryo-ET progresses to resolving small ion channels, integrating optogenetics into our CoVET pipeline will facilitate single-neuron excitation, enabling detailed correlations between electrophysiological properties, such as synaptic connectivity and firing pattern, and molecular structural states. These advancements will significantly enhance our ability to understand how electrophysiological heterogeneity correlates with molecular diversity at the single-cell level.

We have carefully revised our discussion to clearly outline current technical challenges as follows:.

“Discussion

~In terms of molecular phenotypes, while the expression and localization of ion channels are closely related to heterogeneous neuronal responses^{12,13}, current technical limitations in cryo-ET present significant challenges for visualizing small membrane-bound complexes, such as voltage-gated sodium and potassium channels (~200 kDa) in situ¹⁴. These channels are small, embedded in membranes, and are often sparsely distributed, making them difficult to detect in a crowded intracellular environment. Furthermore, comprehensive structural analysis across neurons, including dendritic compartments, is hindered by the limited material volume in dendrites, which makes it difficult to prepare stable lamellae. The cryo-FIB milling of such thin structures requires low throughput and is technically demanding. Considering these constraints, we focused on the ribosomal structures in the soma, which are larger and more abundant complexes amenable to current cryo-ET techniques. Although ribosomes are not directly responsible for electrical activity, their role in activity-dependent translation provides an informative molecular readout of neuronal function⁹. This focus enabled us to capitalize on the strengths of cryo-ET while probing molecular variations across different electrophysiological states. As imaging technologies advance, enabling the reliable visualization of smaller complexes and finer structures, our CoVET pipeline will be well-

positioned to elucidate the structural basis of neuronal responsiveness with high molecular precision. ~”

QI-2. The observed differences in the translational profiles of ribosomes are more likely a consequence of electrical stimulation, rather than a direct correlation with the intrinsic electrophysiological state. As the authors noted, electrical activity can induce translation. It's reasonable to expect that the responsive neurons were more strongly perturbed by the electrical stimulation and consequently triggered changes of the translational profiles in them.

RI-2. We appreciate the reviewer's insightful comment and fully agree that the observed differences in ribosomal translational profiles could reflect responses to electrical stimulation rather than purely intrinsic electrophysiological states. Indeed, previous studies have demonstrated that neuronal electrical activity can induce changes in translation by biochemical profiling.^{15,16}

To directly examine whether electric field stimulation significantly contributed to the observed translational heterogeneity, we conducted additional control experiments. Specifically, we compared the translational landscapes of unstimulated neurons (control grid) with those of neurons subjected to electric field stimulation, using cultures prepared from the same animal (Fig. RI-2a-c). We analyzed 6 unstimulated neurons (with a total of 9,562 ribosomes) and 7 stimulated neurons (with a total of 7,367 ribosomes). While both groups of neurons exhibited heterogeneity in their translational landscapes, neurons exposed to electrical stimulation displayed slightly increased heterogeneity compared to unstimulated neurons (Fig. RI-2c). These results support the reviewer's suggestion that the observed differences in ribosomal translational profiles likely include the effect arising from electrical stimulation.

While there are differences in heterogeneity from both groups, it is not dramatically significant, considering the statistical significance. There are several possible factors influencing this result. Previous studies validated activity dependent transcription and translation utilizing strong and consistent stimulation, such as high concentration of KCl, glycine and high frequency optogenetic stimulation, compared to our weak and brief stimulation (2Hz, 20 s (applied stimulation duration), < 15 min (total imaging duration before cryo-fixation))^{9,15,17,18}. Therefore, weak and brief stimulation in our experimental scheme may not influence the translational landscape significantly. Furthermore, another previous study revealed that neurons which are before fully synchronization exhibit notably heterogeneous spontaneous activity⁷. In our experimental scheme, because we utilized DIV 10 ~ 14 neurons which were before fully synchronized, heterogeneous spontaneous activity across the neurons before voltage imaging would be present and this might be reflected in neuronal responsiveness upon stimulation as we observed. Therefore, the heterogeneous spontaneous activity during their growth could shape the global translational landscapes via activity-dependent translation manner before stimulation. To find the crucial factor influencing the translational landscapes, we should perform orthogonal biochemical assay and far larger cryo-ET analysis. While the biological validation is important, because our primary goal of this manuscript is technical advancements in cryo-ET, we have refrained from performing these experiments.

Therefore, we cannot attribute the heterogeneity in translational landscapes to any single factor, so rather than enumerating every possible cause in the main text, we have toned down

our biological interpretation and simply present the observed differences among the clusters.

Figure RI-2. Translational landscapes of neurons **a**, neurons without electric field stimulation. (n=6) **b**, neurons with electric field stimulation. (n=7) **c**, bar plot of standard deviation between two groups. To validate the statistical significance of variation, Fligner-Killeen’s test was performed. P value under 0.05 was indicated on the plot.

QI-3. The term “AUC” is commonly associated with ROC curve in classification. I would suggest using a different term to avoid confusion.

RI-3. Thank you for pointing out the confusing terminology in our manuscript. As we mentioned above, we decided not to use AUC because it is not an orthogonal parameter to the peak value. Instead of using AUC, we used the ‘Decay parameter’ which indicates the degree of decrease in membrane potential during a repolarization phase.

QI-4. I cannot find any description regarding the measure of “reproducibility”

Thank you for pointing out our mistakes. We have added the methods to measure reproducibility in the methods section.

Reviewer #2 (Remarks to the Author):

Summary

This study represents the first example to my knowledge of electrophysical/chemical information being used to study cells in a correlative manner for study using cryo electron tomography. The ability to use this information is novel and represents a significant addition to the field.

The authors further used this information to inform the latent translational profiles of pre-characterised cells, leading to low resolution structures of ribosomes in different stages of translation which the authors could then link to the electrophysical profile of the specific cell. This demonstrates that the technique could be used to unpick structural details using electrophysiology as a selection criterion.

The ribosomes studies, while interesting, are nevertheless at a relatively low resolution compared to other studies on ribosome translational dynamics and the information gained on the structures is limited to sub-unit level insight. This is not a critical consideration for the work, however.

In general, this study is worthy of merit and could be used as a basis to study various other neuronal phenomena where electrophysiology has a direct impact on cellular function. I really enjoyed the work and especially the concept behind it.

We thank the Reviewer for being enthusiastic in our study and for recognizing its significance. We have revised the manuscript to address all the points raised in the review, as described below.

Major Comment

QII-1 The major change to this manuscript should be in the “Preparing primary neuron for effective CoVET analysis” section. The use of co-cultures to study neurons is not novel, including where the cultures are to be used for cryo electron tomography. See 10.1126/science.1261197 for an example. It would therefore be useful for the authors to tone down the novelty of this aspect in the results and focus more specifically on the need to grow cells at low density for imaging and incorporate this into the results for the need to design a new sandwich assembly.

RII-1 Thank you for your valuable feedback on our manuscript. We acknowledge that the use of co-cultures, including their application in cryo-electron tomography, is not novel, as exemplified by the study you referenced (10.1126/science.1261197).

We have revised the section to moderate claims regarding the novelty of the co-culture method and have appropriately cited the referenced study. Additionally, we have clarified the advantages of our system, particularly its ability to support low-density cell growth for imaging, and incorporated this into the results to highlight the necessity of designing a new sandwich assembly. To further emphasize these improvements, we have included additional figures (Fig. 3a, b, Fig. RII-1) demonstrating the system’s compatibility with live-cell imaging devices,

which are essential for accurate voltage imaging. These revisions emphasize the methodological integration of our system with live-cell imaging, ensuring its effectiveness in voltage imaging applications.

“Results

~However, high-density cultures lead to clumping of neurons, making it difficult to visually identify single neurons, which is crucial for optical analysis. To achieve a low-density culture on grids without compromising neuronal health, we employed a sandwich culture¹⁹⁻²¹ on grids to optimize the lower cell population. This ensures normal functional development of neurons while maintaining a sufficiently low density to identify individual neurons for both cryo-ET and voltage imaging.~”

Fig. RII-1. Compatibility between live cell imaging chamber and grid holders. **a.** overall architecture of grid holder and imaging chamber on fluorescence microscope. **b.** Detailed scheme of grid holders in imaging chamber with electric stimulator.

Minor Comments

QII-2 The sentence “Given the correlation between electrophysiological properties and molecular society of neurons, single-cell analysis techniques based on electrophysiological properties, such as Patch-seq and Voltage-seq^{11,12}, are emerging as essential tools to characterize dynamic molecular changes to the heterogenous electrophysiological property” should be reworded to remove molecular society and remove the reference to electrophysiology being essential tools to characterise molecular details. It does not

RII-2 Thank you for your careful reading of our manuscript and for pointing this out. In response, we have removed the sentence to avoid the use of the term "molecular society" and to eliminate any implication that electrophysiology is essential for characterizing gene expression. As there are no direct references supporting the importance of structural characterization

based on electrophysiological properties, we now present its significance as follows:

“Introduction:

~The spatial distribution and structural heterogeneity of molecular networks influences neural functions through neural plasticity, ranging from synaptic ultrastructures to network-level connectivity^{22,23}. Disruption of these molecular networks results in abnormal neural activity, leading to impaired signal transduction. These disruptions are associated with severe phenotypes, including aging, cognitive decline, and neurodegeneration^{24,25}. Thus, there is considerable demand for structural characterization of spatial molecular networks within electrophysiological properties. ”

QII-3 Multiple references/uses of phrase “molecular society” should be re-worded as it is ambiguous in meaning.

RII-3 Thank you for your feedback regarding the use of the phrase "molecular society." We recognize that this term may be ambiguous, and we have revised multiple instances throughout the manuscript to use more precise language that clearly conveys the intended meaning. Where applicable, we have replaced it with specific references to molecular networks, macromolecular complexes, or cellular architecture, depending on the context.

QII-4 There is no need to always capitalised the first letters of acronyms when written out i.e. In situ cryo-Electron Tomography (cryo-ET). There are several instances of this.

RII-4 Thank you for your careful review. We have corrected instances where acronyms were unnecessarily capitalized when written out, ensuring consistency with standard scientific conventions. We have carefully reviewed the manuscript to apply this correction throughout such as “cryo-electron tomography”, “cryo-correlative light and electron microscopy”, “cryo-focused ion beam/scanning electron microscopy”

QII-5 “samples culture method” the word method is redundant.

RII-5 Thank you for your comments for the redundancy of words. We have revised “sandwich culture method” to “sandwich culture”.

QII-6 “One day prior to seeding neurons, a grid was mounted onto the grid holder” Is this a necessary step?

RII-6 Thank you for your thoughtful question. This step is necessary. Grids are fragile and difficult to handle in liquid, so we pre-mount them onto the holder to ensure stability during coating and subsequent processing. To promote reliable neuronal attachment, grids are coated with poly-D-lysine and incubated for an extended period. Pre-assembling the grid with the holder in advance allows for this coating process and helps maintain reproducibility and structural integrity throughout the CoVET workflow.

QII-7 Figure 1: Might be useful to restructure/alter the figure to make it clearer that the neuronal

activity characterisation is done elsewhere, analysed and then the results incorporated into the correlation.

R11-7. Thank you for your comments about clarifying the overall scheme of our workflow. We acknowledge the ambiguity in figure 1 as you commented and revised the figure 1 accordingly (Fig. R11-7).

Fig.R11-7. Overall schematic diagram of CoVET

Q11-8 “We demonstrated our combined pipeline by exploring the translational landscapes of ribosomes within an electrophysiological context.” The use of electrophysiological context is odd and should be reworded.

R11-8 Thank you for your comments about the phrase we misused. We revised the sentence to “We demonstrated our combined pipeline by exploring the translational landscapes of ribosomes in relation to neuronal responsiveness.”

QII-9 As I touched on in the summary, the resolution of the ribosome structure generated is lower than similar studies investigating in-situ ribosomes (DOI: 10.1038/s41592-020-01054-7; 10.1038/s41467-023-36372-9; 10.1126/science.adh1411). Some of these studies use fewer particles with equivalent equipment and pixel sizes yet achieve a higher resolution. Can the authors comment on why they were unable to replicate these published results? With ~20000 particles I would expect a better resolution for the global structure.

RII-9 Thank you for your thoughtful feedback regarding the resolution of our ribosome structure. We acknowledge that resolution is influenced by multiple factors, including the use of FIB milling, the type of ion beam (e.g., gallium or plasma) used for milling, and pixel size²⁶. Among the reference studies, the most comparable condition to our system which utilized gallium ion beam milling, achieving 6.7 Å resolution with 28K particles at 2.17 Å/pix, which corresponds to 3.1 folds of pixel size²⁷.

We therefore reasoned that the discrepancy in resolution may arise from factors in our cryo-ET image processing. To address this, we reanalyzed our data using 31K ribosomal particles, which improved the consensus map resolution to 7.8 Å at 2.42 Å/pixel, which corresponds to 3.2 folds of the pixel size. This global resolution is comparable to that of the reference study.

Moreover, our primary goal was to classify the global translational landscapes of ribosomes and compare these across functional clusters. At this level of analysis, our achieved resolution was sufficient to distinguish between different translational states.

QII-10 Can the authors prove an FSC curve?

RII-10 We have now included the FSC curves for the consensus and each state map of subtomogram averaging of cryoET in Supplementary Figure 3 (Fig. RII-10).

Fig.RII-10. FSC curve of ribosomes

QII-11 Will the authors deposit the data on EMPIAR?

RII-11 We appreciate the reviewer's suggestion. We have deposited the data in EMPIAR under the accession code EMPIAR-12606 and have included this information in the Data Availability section.

QII-12 "greatly simplifying the workflow and making CoVET universally accessible to the cryo-ET applications" This sentence should be reworded and an explanation of what the simplification step is.

RII-12 We appreciate the reviewer's suggestion. We have reworded the sentence to clarify the simplification step:

"~By employing a chemical voltage indicator and providing broad compatibility with conventional live-cell chambers, our approach simplifies workflow and enables the measurement of responsiveness in all neurons without the need for genetic modification, making CoVET widely accessible for cryo-ET applications.~"

QII-13 “Moreover, integrating CoVET with conventional cryo-CLEM¹⁷ adds multiple layers of information, enabling precise identification of the regions of interest within the target neurons.” This sentence should be re-written as it is not clear what the multiple layers of information are, or is the precision of identification of regions using this approach characterised.

RII-13 We appreciate the reviewer’s suggestion. We have revised the sentence for clarity:

“Integrating CoVET with conventional cryo-CLEM enables precise identification of regions of interest by combining fluorescence-based protein localization with characterization of individual neuronal responses.”

QII-14 “By using CoVET to pinpoint neurons with electrophysiological defects and then performing in situ structural analysis, we can directly correlate functional abnormalities with structural phenotypes, such as amyloid beta or tau aggregation¹⁴.” This sentence should be either removed or changed as the claim to be able to use this to target Abeta or Tau is unproven – speculation over applications should be more clearly worded.

RII-14 We appreciate the reviewer’s feedback and have removed the speculative sentence from the *Discussion* section.

QII-15 “Ribosomal landscapes” What does that mean?

RII-15 Thank you for pointing out the phrases we misused. We aimed to describe the heterogeneity of ribosomal conformations. To improve clarity, we have revised the terminology with ribosomal conformations or translational landscapes depending on the context.

QII-16 The manuscript should be proofread and grammatical errors corrected.

RII-16 We conducted a thorough, line-by-line revision of our manuscript to ensure accurate terminology throughout. Additionally, we engaged a professional scientific editing service to further refine the language and enhance clarity. We are grateful for the reviewer’s careful attention to these details and trust that our revisions adequately address the concerns raised.

Reviewer #3 (Remarks to the Author):

Jung et al. developed a technique they call Correlative Voltage Imaging and cryo-ET (CoVET). They combined voltage imaging of cultured neurons with subsequent FIB milling and cryo-electron tomography. The authors designed specialized neuron culture and imaging methods that enable mapping between real-time voltage imaging traces and in situ cryo-ET images. They report significant variations in ribosome states among neurons with differing excitability.

The ability to correlate physiology with ultrastructure is an important goal, and this paper provides a technical advance in that direction. However, the description of the technology is insufficiently detailed, the statistics are weak, and the biological claims are unconvincing. A manuscript focused on the technical aspects of the measurements that avoids making unconvincing biological claims could be suitable for publication.

We appreciate the reviewer's thoughtful feedback and have revised our manuscript accordingly. To enhance clarity, we have incorporated key methodological details directly into the main text, providing a more comprehensive description of the CoVET technique, including neuron culture optimization, the voltage imaging setup, and the correlation workflow with cryo-ET. Additionally, we have strengthened our statistical analyses by increasing the number of biological replicates, performing multiple hypothesis testing with appropriate post-hoc analyses to improve the rigor and transparency of our comparisons across electrophysiological clusters. Regarding the biological interpretation, we acknowledge the need for caution and have revised the discussion to focus primarily on the methodological advancements of CoVET rather than making strong biological claims. Below, we provide a detailed, point-by-point response to each of the reviewer's comments.

1. Many of the key details were left for the Methods, but since this is a paper specifically about a new method, it would make sense for all the non-standard aspects to be in the main text. Here is some information that is in the Methods which should be in the main text.

Thank you for your insightful suggestion. We agree that, given the methodological focus of our study, key non-standard aspects should be clearly presented in the main text. In response, we have improved Figure 1 for the overall workflow and incorporated essential methodological details from the Methods section into the main text, including specifics on neuron culture optimization, voltage imaging setup, the custom grid holder, and the workflow for correlating electrophysiological data with cryo-ET.

QIII-1-1. Please state the composition of the TEM grid: what was the surface material, and what were the support bars made of? Was it cleaned or treated prior to addition of PDL and Laminin?

RIII-1-1. We used a gold grid coated with carbon film. Prior to poly-D-lysine (PDL) and Laminin coating, the grids were glow-discharged and sterilized with UV for 1 hour. We have now elaborated on these details in the Results section

“RESULTS

~Gold grids coated with carbon film were glow-discharged and UV-sterilized prior to coating with poly-D-lysine (PDL) and laminin. After sterilization, the grids were packaged into sterilized grid holders and coated with PDL.~”

QIII-1-2. Please specify the seeded cell density (cells/cm²) rather than saying just “one-fifth of the conventional neuronal culture,” since different labs have different conventions.

RIII-1-2. We appreciate your suggestion and have revised our manuscript accordingly. The text now specifies the cell density we used: *“~ 45,000 cells/cm² were seeded onto the packaged grids.”*

QIII-1-3-i. Please explain how the coverslip was attached to the back of the grid and the grid holder.

RIII-1-3-i. Thank you for your comments and we added the details in the result section:

“Results

~After mounting the grid to the grid holder, vacuum grease was applied using a paintbrush at 3 different spots along the backside of the grid holder(to-be interface between the grid holder and coverslip). A 18 mm round coverslip was firmly attached by pressing it against the backside of the grid holder.~”

QIII-1-3-ii. How were the paraffin dots produced and applied to the bottom coverslip?

RIII-1-3-ii. We added the sentence in the result section:

“Results

~To achieve appropriate spacing between the feeder layer and the packaged grid, commercially available flat-bottomed paraffin wax dot was used. Three paraffin wax dots were slightly melted on the bottom and allowed to attach to the 12-well plate bottom”.

Additionally we revised the method section for more detail.

QIII-1-4. What were the nuclei-specific fluorescence markers? When and how were these delivered?

RIII-1-4. The neurons were labeled with Hoechst33258 together with BeRST. This information is included in the Results section as follows:

“Results

To characterize the electrophysiological properties of individual neurons, cultured cells were

preincubated with the voltage-sensitive probe BeRST1, and DNA labeling probe Hoechst was applied concurrently to label nuclei for further analysis.”

QIII-1-5. It would be helpful to have a figure describing the cryoET workflow too, since for non-experts this seems quite complex.

We included the workflow of cryo-ET in the overall scheme of CoVET (Fig. 1, Fig. RIII-1-5)

Fig. RIII-1-5. Overall schematic diagram of CoVET

2. Statistics:

QIII-2-1. Fig. 1d) How many biological replicates, i.e. independent plating rounds, was this? There can be substantial batch-to-batch variability in neuron culture survival, so this needs to be tested across replicate experiments.

RIII-2-1. Thank you for your feedback regarding the statistical analysis. In the initial manuscript,

we performed Sholl analysis on only one grid per condition (with and without an astrocyte feeder layer). As you pointed out, this low number of biological replicates undermines the reliability of our conclusions. To address this, we conducted additional experiments across multiple grids. Specifically, we performed Sholl analysis (Fig. 2d) on 10 grids per condition, using neurons from three different rats on three separate days. With these expanded replicates, we observed statistically significant differences between the two groups (Fig. RIII-2-1). These results demonstrate that low-density neuronal culture on EM grids—a prerequisite for effective CoVET—cannot be achieved in the absence of an astrocyte feeder layer.

Fig. RIII-2-1. evaluation of dendritic development. Number of neurites in 20 µm from each neuron soma was counted according to DIV in both groups cultured with a feeder layer (n = 724, 510, 398, 374, 350) and without feeder layer (n = 632, 280, 204, 161, 150) from 10 different grids each. Data were shown as mean ± s.d. Unpaired t-tests were performed to test statistical significance. * $P < 0.05$, *** $P < 0.001$, **** $P < 0.0001$

QIII-2-2. Fig. S2 is a crucial context on the batch-to-batch variability and should be in the main text.

RIII-2-2. Thank you for emphasizing the importance of batch-to-batch variability in the electrophysiological response parameters. To address this point, we analyzed data from four independently prepared grids across four separate days. As shown in Supplementary Fig. 2, we observed substantial variability in each parameter across batches. These results highlighted the significant heterogeneity of neuronal responses even under standardized conditions. To reflect the significance of this variability, now we included these results in the main figure 3i.

Fig. RIII-2-2. Batch by batch heterogeneity in parameters of neuronal response, Peak value(left panel), Decay parameter (middle panel), Reproducibility (right panel). One-way ANOVA followed by Tukey's post hoc test for multiple comparisons was performed to assess statistical significance. * $P < 0.05$, *** $P < 0.001$, **** $P < 0.0001$.

QIII-2-3. The clustering into 3 classes is not sufficiently justified. Does the distribution of ephys properties actually fall into three distinct groups, or is it a broad distribution which is being divided up? The authors should plot scatter plots of the neurons for all pairwise combinations of parameters to determine whether there are really 3 clusters.

RIII-2-3. We appreciate the reviewer's insightful comment regarding the justification of three electrophysiological clusters. As noted, the original parameters—Peak Value, Area Under the Curve (AUC), and Reproducibility—showed substantial interdependence. Specifically, we found strong correlations among these parameters: Peak Value vs. AUC ($R^2 = 0.63$), AUC vs. Reproducibility ($R^2 = 0.57$), and Peak Value vs. Reproducibility ($R^2 = 0.85$), suggesting that they may not provide fully independent axes for clustering (Fig. R3-2-3a).

To address this, we introduced a new parameter—*decay parameter*—which reflects the decay rate of the voltage response after stimulation, thereby capturing a distinct dynamic property of the neuronal response^{28,29}. Incorporating decay parameter reduced the overall correlations: Peak Value vs. Decay parameter ($R^2 = 0.50$), Decay parameter vs. Reproducibility ($R^2 = 0.40$), while Peak Value vs. Reproducibility remained high ($R^2 = 0.85$) (Fig. R3-2-3b). Despite this residual correlation, we retained Reproducibility to account for variability in sporadic responses, particularly relevant for the intermediate response group.

We then plotted pairwise scatter plots of all parameters (Fig. R3-2-3c). These revealed a broad distribution with no clearly defined separations, supporting the reviewer's concern. However, when we attempted clustering with only two categories—responsive and non-responsive—we consistently observed an intermediate group with inconsistent but detectable responses. This motivated the inclusion of the “moderately responsive” cluster, which captures biologically relevant heterogeneity in response strength and reliability. Therefore, while the overall distribution is continuous, our three-cluster model reflects empirical distinctions in response behavior and improves interpretability of downstream structural analysis.

Fig. RIII-2-3. Pairwise plot between parameters of neuronal response. **a**, Pairwise plot between parameters based on clustering utilizing AUC parameter **b**, Pairwise plot between parameters based on clustering utilizing Decay parameter.

QIII-2-4. p. 9: in the statistics of neurons found and neurons lost during vitrification, how many independent grids was this procedure performed on?

RIII-2-4. Thank you for your careful review. In the initial manuscript, we assessed neuronal attachment on three grids. We have performed an additional experiment in a follow-up experiment and acquired data from a total 4 different grids. As a result, we confirmed that, across all four grids, $92.6\% \pm 6.4\%$ (95% confidence interval) of neurons remained stably attached after vitrification. We revised the manuscript accordingly.

Results

"In this process, we could identify 92.6% of neurons whose electrophysiological characteristics were measured. Average ~7% of neurons from 4 different grids were lost during transfer and vitrification handling."

QIII-2-5. Please give statistics on the numbers of neurons subjected to cryoET in each cluster, and the distribution of numbers of ribosomes per neuron. How many biological replicates (independent platings and measurements) were the data recorded from? Ah, the information is buried in the supplement: The whole paper is based on measurements on groups of 4, 3, and 5 neurons from the three clusters, respectively (presumably from a single grid?). In this case, all of the bar graphs should indicate the specific individual values for each neuron. This is unlikely to be enough cells to make any firm biological conclusions about the relation between the ephys properties and the ribosome properties.

RIII-2-5. We appreciate the reviewer's careful attention to the sample size and its implications

for biological interpretation. In response to your comment, we conducted additional experiments and expanded our dataset to include 19 neurons, derived from 4 independent grids—each grid prepared on a different day using primary neurons from different animals. Based on our clustering analysis, these neurons were classified as follows: Cluster 1 (strongly responsive), $n = 5$ neurons; Cluster 2 (moderately responsive), $n = 6$ neurons; Cluster 3 (non-responsive), $n = 8$ neurons. The data shown in Fig. 5d represents the averaged portions of each conformation from each cluster, with error bars indicating the SEM calculated from all cells within each cluster.

We fully agree with the reviewer that the current sample size limits the strength of our biological conclusions. As such, we have revised the manuscript text to tone down our claims, reframing them as preliminary observations that suggest electrophysiology- correlated differences in translational landscapes, rather than definitive biological conclusions.

Importantly, we note that such sample sizes, while limited, are currently common in cryo-ET studies due to the technical demands of cryo-FIB milling, cryo-ET data acquisition, and downstream processing. These steps remain time- and resource-intensive, but we are optimistic that ongoing advances in automation and throughput will soon enable more statistically powered studies. In that context, we see the CoVET platform as a foundational tool that can facilitate deeper insights into the structural basis of neuronal heterogeneity.

QIII-2-6. For the mapping of ribosome conformations onto ephys states, were the p-values corrected for multiple-hypothesis testing? They should be. Was the n value used for the number of ribosomes or the number of cells?

RIII-2-6. We thank the reviewer for this important point regarding statistical analysis. We fully agree that utilizing the number of cells as the unit of replication, along with multiple hypothesis testing, better accounts for biological variability and avoids artificially inflating statistical power through particle counts. Therefore, to address the issue of multiple hypothesis testing, we have revised our statistical analysis by applying one-way ANOVA followed by Tukey's post-hoc test for all comparisons of ribosomal conformational distributions across electrophysiological clusters. This approach appropriately accounts for multiple pairwise comparisons and controls the family-wise error rate. The updated significance levels are indicated in Fig. RIII-2-6, where asterisks denote statistically significant differences between groups (* $P < 0.05$, ** $P < 0.01$, *** $P < 0.001$).

With the addition of new biological replicates, the statistical robustness of our comparisons has improved, and key differences—particularly in the decoding1 state (Fig. RIII-2-6a, b) and rotated2 state (Fig. RIII-2-6b)—remain statistically significant after correction under our experimental conditions.

Fig. RIII-2-6. Translational landscapes. **a**, translational landscapes of total ribosomes **b**, Translational landscapes of polysome. Data are presented as mean \pm S.E.M. One-way ANOVA followed by Tukey's post hoc test for multiple comparisons was performed to assess statistical significance. (Biological replicates: Cluster1: 5, Cluster2: 6, Cluster3: 8) * $P < 0.05$, ** $P < 0.01$, *** $P < 0.001$.

QIII-3. Electric field stimulation: I'm very confused about the electric field geometry. It looks like the stim electrodes are right above the stainless steel disk, in which case the disk will shunt most of the electric fields around the neurons. Furthermore, if the neurons are plated on a conducting grid, the grid would further shunt most of the electric field. The authors must provide a clearer view (e.g. a cross-section of the geometry during stimulation) and should provide 3-D finite-element simulations to model the electric field.

RIII-3. We thank the reviewer for raising this important point regarding the geometry and potential field-shunting effects of the conductive components in our stimulation setup. We agree that under electrostatic conditions, conductors such as the stainless steel holder and gold EM grid would indeed shunt the electric field due to Gauss's law. However, in our experiments, biphasic square-wave electric pulses were applied, which create time-varying electric fields governed by Maxwell-Faraday's law rather than static conditions.

Under these electrodynamic conditions, a time-varying magnetic field can induce circulating electric fields, even within conductive materials. Moreover, the physical configuration of our stimulation system, specifically the perfusion chamber geometry, electrode placement, grid holder shape, and insulating spacers, was likely to minimize current shorting across the stainless steel and to permit sufficient field exposure to the neuronal layer.

To address the reviewer's concern directly, we have added a cross-sectional schematic diagram (Fig. RIII-3a) illustrating the spatial configuration of the electric field stimulation setup, including the electrode placement relative to the grid, stainless steel holder, and neuron layer. Additionally, we performed 3D finite-element simulations using COMSOL Multiphysics (Fig. RIII-3b, Fig. 3c) to model the electric field distribution during stimulation. These simulations incorporate the spatial geometry of the system, material properties (conductive and dielectric), and stimulation parameters (1 ms biphasic square pulses at 100 mA, saline conductivity ~ 1.5 S/m). As a result, averaged 1089 V/m of electric field was applied on the cultured neuron,

which is sufficient to evoke action potential (Fig. RIII-3c)⁵. These findings are consistent with our experimental voltage imaging results and support the functional validity of the stimulation approach. Relevant figures and a discussion of the stimulation mechanism have been added to the revised manuscript.

“Results

~To validate that our electric field stimulation system with a grid holder can deliver adequate stimulation, we performed electric field simulations (Fig. 3c). Our simulation indicated that, on average, a 1,089 V/m electric field was applied to the neurons on the grids, which is sufficient to evoke action potentials⁵.~”

Fig. RIII-3. Results of electric field simulation. **a**, Schematic diagram of architecture of electric field stimulator with grids **b**, Overall applied electric field stimulation. Upper panel indicates the applied electric field within the whole chamber. Lower panel indicates a magnified view of the applied electric field within the grid holder. **c**, Applied electric field to neurons on grids.

4. Biology: The authors do not sufficiently explain on the significance of the observed three groups of neurons. Rat hippocampal cultures are composed of heterogeneous neuron types, each exhibiting distinct electrical behaviors. Can authors identify the major factors behind these groups? Are they associated with differences in neuron types, maturation states, morphology, or health conditions? The absence of such information weakens the interpretation of the subsequent characterization of ribosome states. Here are also suggestions on experiment details:

Thank you for your insightful comments. Regarding neuronal subtype, we found that although excitatory and inhibitory neurons differ in their ability to follow the frequency of electric field stimulation, both types reliably fire an action potential in response to individual electric field stimulation³⁰. In terms of morphology, the electrophysiological properties and detailed structures of axon and dendrites are strongly correlated and classification based on combination both properties allow the reliable sub-type classifications which display a distinct gene expression profile². Despite the importance of morphological property, because we used low magnification for fast whole grid screening to ensure neuronal health in our experimental set-up, the spatial resolution is insufficient to characterize fine detail of their structure. Nonetheless, minimally, we analyzed soma size and local cell density around each neuron but observed no correlation with responsiveness (Fig. RIII-4).

At the same time, neuronal maturation and overall health are known to affect neuronal activity profile⁶. Indeed, because we performed our analyses at 10-14 days in vitro, there is heterogeneity in maturity across the neurons. These factors could be reflected in neuronal

responsiveness.

To characterize morphological details and maturity of individual neurons without prolonged imaging before cryo-fixation which negatively affect the cellular integrity, development of post-hoc fluorescence analysis compatible with cryo-ET is necessary. Combining the post-hoc analysis with CoVET enables more precise interpretation of link between electrophysiological properties and molecular structures.

We included the discussion about the heterogeneity in neuronal response in Discussion section as follow:

“Discussion

~A variety of factors, including morphological characteristics and neuronal maturation, are thought to contribute to electrophysiological heterogeneity^{1,6}. Prior studies have demonstrated strong correlations between electrophysiological properties and detailed morphological features, showing that the integration of these parameters enables the reliable classification of neuronal subtypes with distinct transcriptomic profiles^{2,3}. Moreover, neuronal maturity has been shown to strongly influence electrophysiological properties^{6,7}, and uneven maturation across the population further broadens the distribution of neuronal responses. In the present study, we observed substantial variability in neuronal responses, both across individual cells and between different culture batches (Figure 3f). While CoVET enables the correlation of electrophysiological properties with in situ molecular architecture, the absence of complementary biological information at single cell level, such as detailed morphological features and neuronal maturity, may limit the ability to accurately interpret these correlations. Without such contextual information, the heterogeneity among neurons may obscure underlying biological realities, making it difficult to dissect functional relationships at the molecular level. Since we employed a rapid whole-grid screening strategy at low magnification to preserve cellular viability in this study, our imaging inherently lacked the spatial resolution required to resolve fine morphological features such as axonal and dendritic structures. To fully integrate CoVET analysis with detailed morphological characterization and measurements of maturity without prolonged imaging before cryo-fixation, the development of post-hoc fluorescence analysis compatible with cryo-ET is necessary. Incorporation of post-hoc analysis with CoVET would enable retrospective morphological characterization and measurements of maturity, providing a more comprehensive framework for correlating electrophysiological properties with in situ molecular structures.~”

Fig. RIII-4. Pairwise plot between response parameters and morphological parameters

QIII-4-1. Differences in spike height are unexpected, as triggered action potentials should be consistent. The authors should rule out the possibility that this variability reflects the role of background autofluorescence..

RIII-4-1. Thank you for this important point. You are correct that individual action potentials have highly consistent depolarization amplitudes. In our study, however, we report the “peak height” as the maximum integrated signal within each 10 ms time bin—effectively reflecting firing rate rather than single-spike amplitude. Because action potentials occur on a ~1 ms timescale, multiple spikes can fall into a single 10 ms bin, producing variability in the measured peak heights.

To confirm this, we performed voltage imaging at 400 Hz (2.5 ms bins) and then down-sampled the data to 100, 40, 10 Hz (Fig. RIII-4-1). The high-speed trace shows uniform peak amplitudes for individual spikes, whereas the down-sampled trace recapitulates the variability in peak height. However, due to the readout speed of the camera, further increasing frame rate forces a reduction in field of view. Consequently, this makes prolonged imaging necessary for whole grid screening and that could negatively affect cellular health due to the phototoxicity. Thus, we settled on 100 Hz as the optimal compromise between temporal resolution, spatial coverage, and cell health.

As you note, our imaging system can be reconfigured to higher frame rates over smaller fields of view when precise spike-by-spike voltage dynamics are required for a few identified neurons and corresponding structural analysis depending on the purpose of the experiment.

Fig. RIII-4-1. Voltage trace in various temporal resolution

QIII-4-2. Peak height, Area Under the Curve (AUC), and reproducibility appear to be strongly correlated, as shown in Supplementary Figure 2. It is not clear if these two parameters give independent information.

RIII-4-2. Thank you for this important observation. We indeed observed strong correlations among the original parameters, Peak Height, AUC, and Reproducibility, particularly between Peak Height and Reproducibility. To improve independence among features, we replaced AUC with “Decay parameter”, which reduced overall correlation across parameters. Despite this, peak value and reproducibility remained correlated, yet they capture distinct aspects of neuronal responsiveness: while peak value reflects the magnitude of depolarization, reproducibility captures the consistency of response across multiple stimulations. Some neurons exhibited sporadic firing with high peaks but poor reproducibility, while others responded consistently with lower peaks. Therefore, neither metric alone was sufficient to describe the response heterogeneity. By using both peak value and reproducibility as complementary parameters, we aimed to more represent the diverse response profiles of individual neurons.

QIII-4-3. When characterizing intrinsic excitability, synaptic blockers are needed to block communication between neurons. Imaged neurons could be silenced or excited by their neighbors.

RIII-4-3. We appreciate the reviewer’s important point regarding synaptic connectivity. While we did not apply synaptic blockers, our experiments were conducted under low-density culture conditions, where spontaneous activity and recurrent network dynamics were minimal. Nevertheless, we acknowledge that excitatory and inhibitory synaptic inputs could have modulated the temporal features of the electric field–evoked responses. Thus, our measurements likely reflect a composite response shaped by both intrinsic excitability and local synaptic interactions. Rather than isolating intrinsic properties alone, our aim was to capture the functional output state of each neuron within its microenvironment.

QIII-4-4. The authors characterize the distribution of ribosome states at the neuron cluster level. How accurately can the ribosome patterns of individual cells predict their electrical signatures?

RIII-4-4. Thank you for this insightful suggestion. To explore how well ribosome state distributions reflect electrophysiological phenotypes, we performed hierarchical clustering based on the translational landscapes of individual neurons (Fig. RIII-4) and compared the resulting groups to the original electrophysiological clusters using the Adjusted Rand Index (ARI). The ARI was 0.08, indicating low concordance between the two clustering schemes. This suggests that while electrophysiological responsiveness correlates with ribosomal state at the population level, ribosome-based clustering alone is currently insufficient to reliably predict the electrical phenotype of individual neurons.

This likely reflects the fact that electrophysiological responsiveness is a macro-scale phenotype, shaped by the integration of numerous molecular processes beyond translation. Additionally, the electrophysiological clustering was performed on ~600 neurons, whereas the ribosome-based dataset included only 19 cells, comprising ~3% of the total. We anticipate that with increased throughput in cryo-ET and broader sampling of neurons, bottom-up inference from translational landscapes to functional state will become more feasible.

Fig. RIII-4-4 Hierarchical clustering based on translational landscapes

QIII-4-5. Could the authors provide more functional insights into the observed ribosome states? For instance, if channel proteins are being preferentially translated, would it be possible to observe an enrichment of ribosomes or polysomes near the cell membrane, endoplasmic reticulum (ER), or secretion-related organelles? Such spatial correlation could provide stronger evidence linking ribosome states to specific functional outcomes and enhance the biological relevance of the findings.

RIII-4-5. Thank you for your insightful suggestions. Previous studies have shown that chemical perturbation of protein synthesis reshapes the global translational landscape^{27,31}: in untreated cells, most ribosomes occupy decoding states, whereas treatment causes a marked decrease in these states. Our observations exhibit a similar trend, suggesting a potential link between neuronal responses and translational activity. However, we acknowledge that this relationship remains speculative in the absence of direct biochemical validation in this study, such as measurements of translational kinetics or polysome profiling. Accordingly, we have revised the manuscript by minimizing biological speculation and placing greater emphasis on the technical development and demonstration of the CoVET method. Nonetheless, we recognize that validating the relationship between neuronal responsiveness and protein synthesis remains an important objective, and we plan to pursue this through biochemical assays in future studies.

In addition to the overall analysis, we specifically examined the translational landscapes of ER-associated ribosomes to focus on membrane protein translation. ER-associated ribosomes were classified by selecting ribosomes located within 18 nm of the membrane, resulting in 2,373 ER-associated ribosomes out of a total of 32,091 ribosomes. We then reconstructed the structures of the ER-associated ribosomes (Fig. RIII-4a) and analyzed their translational landscapes (Fig. RIII-4b). Overall, the translational landscape trends were preserved among ER-associated ribosomes; however, the statistical significance was weaker (Fig. RIII-4-5), likely due to the relatively small proportion of ER-associated ribosomes (<10% of the total). As the reviewer suggested, improved spatial correlation with the ER could better link ribosomal states to functional outcomes. We anticipate that acquiring a larger dataset in future studies will be critical for a more rigorous analysis of ER-associated ribosomal landscapes and for elucidating the biological significance of the electrophysiological clusters.

Fig. RIII-4-5 Translational landscapes of ER associated ribosomes. **a.** Subtomogram averaging structure of ER-associated ribosome. **b.** Translational landscapes of ER-associated ribosomes. Data are presented as mean \pm S.E.M

Minor

QIII-5. In Main text paragraph 1, please provide more discussion on the importance of structure characterization rather than RNA-based characterization.

RIII-5. We removed the sentences about RNA-seq related methods, but in the case of structural characterization of neurons based on the electrophysiological properties, there is no reference. Thus we simply discuss about it as follow:

“Introduction

~When these molecular networks are disrupted, abnormal neural activity can arise, leading to impaired signal transduction. These disruptions are associated with severe phenotypes, including aging, cognitive decline, and neurodegeneration^{24,25}. Thus, there is a considerable demand for the structural characterization of spatial molecular networks within the electrophysiological properties.~”

QIII-6. Please Provide the full name of TEM (Transmission electron microscopy).

RIII-6. We revised our manuscripts accordingly as follow: “For the setup of CoVET, we designed a customized Transmission electron microscopy (TEM)-grid holder for the co-culture system of neurons and astrocytes, allowing for the optimization of cultured neurons and efficient correlated imaging”

QIII-7. In Analysis of voltage imaging data, it should be “8 decibel of peak signal to noise ratio”.

RIII-7. We revised our manuscripts as follow “ 8 decibel of peak signal to noise ratio”

QIII-8. Overall the English needs improvement (e.g. the authors repeatedly use the word “society” in a way that doesn’t make sense).

RIII-8. Thank you for pointing out the readability of our manuscripts. We carefully revised our terminology throughout the manuscripts and performed scientific editing service. We revised the term “society” into “networks”

Reference

1. Grudt, T. J. & Perl, E. R. Correlations between neuronal morphology and electrophysiological features in the rodent superficial dorsal horn. *J Physiol* **540**, 189–207 (2002).
2. Cadwell, C. R. *et al.* Electrophysiological, transcriptomic and morphologic profiling of single neurons using Patch-seq. *Nat. Biotechnol.* **34**, 199–203 (2016).
3. Muñoz-Manchado, A. B. *et al.* Diversity of Interneurons in the Dorsal Striatum Revealed by Single-Cell RNA Sequencing and PatchSeq. *Cell Rep* **24**, 2179–2190.e7 (2018).
4. Stern, S., Rotem, A., Burnishev, Y., Weinreb, E. & Moses, E. External Excitation of Neurons Using Electric and Magnetic Fields in One- and Two-dimensional Cultures. *J Vis Exp* (2017) doi:10.3791/54357.
5. Radman, T., Ramos, R. L., Brumberg, J. C. & Bikson, M. Role of cortical cell type and morphology in subthreshold and suprathreshold uniform electric field stimulation in vitro. *Brain Stimul* **2**, 215–28, 228.e1–3 (2009).
6. Biffi, E., Regalia, G., Menegon, A., Ferrigno, G. & Pedrocchi, A. The influence of neuronal density and maturation on network activity of hippocampal cell cultures: a methodological study. *PLoS One* **8**, e83899 (2013).
7. Cabrera-Garcia, D. *et al.* Early prediction of developing spontaneous activity in cultured neuronal networks. *Scientific Reports* **11**, 1–13 (2021).
8. Held, R. G. *et al.* In-Situ Structure and Topography of AMPA Receptor Scaffolding Complexes Visualized by CryoET. *bioRxiv* 2024.10.19.619226 (2024) doi:10.1101/2024.10.19.619226.
9. Hacısuleyman, E. *et al.* Neuronal activity rapidly reprograms dendritic translation via eIF4G2:uORF binding. *Nat. Neurosci.* **27**, 822–835 (2024).
10. Biever, A. *et al.* Monosomes actively translate synaptic mRNAs in neuronal processes. *Science* **367**, (2020).

11. Held, R. G., Liang, J. & Brunger, A. T. Nanoscale architecture of synaptic vesicles and scaffolding complexes revealed by cryo-electron tomography. *Proc Natl Acad Sci U S A* **121**, e2403136121 (2024).
12. Mondal, Y., Calabrese, R. L. & Marder, E. Activity-Dependent Changes in Ion Channel Voltage-Dependence Influence the Activity Patterns Targeted by Neurons. *bioRxiv* 2025.02.05.636744 (2025) doi:10.1101/2025.02.05.636744.
13. O'Leary, T., Williams, A. H., Caplan, J. S. & Marder, E. Correlations in ion channel expression emerge from homeostatic tuning rules. *Proceedings of the National Academy of Sciences* **110**, E2645–E2654 (2013).
14. Turk, M. & Baumeister, W. The promise and the challenges of cryo-electron tomography. *FEBS Lett* **594**, 3243–3261 (2020).
15. Lituma, P. J., Singer, R. H., Das, S. & Castillo, P. E. Real-time imaging of Arc/Arg3.1 transcription ex vivo reveals input-specific immediate early gene dynamics. *Proceedings of the National Academy of Sciences* **119**, e2123373119 (2022).
16. Tsang, B. *et al.* Phosphoregulated FMRP phase separation models activity-dependent translation through bidirectional control of mRNA granule formation. *Proceedings of the National Academy of Sciences* **116**, 4218–4227 (2019).
17. Buxbaum, A. R., Wu, B. & Singer, R. H. Single β -actin mRNA detection in neurons reveals a mechanism for regulating its translatability. *Science* **343**, 419–422 (2014).
18. Tyssowski, K. M. *et al.* Different Neuronal Activity Patterns Induce Different Gene Expression Programs. *Neuron* **98**, 530–546.e11 (2018).
19. Kaech, S. & Banker, G. Culturing hippocampal neurons. *Nat. Protoc.* **1**, 2406–2415 (2007).
20. Roppongi, R. T., Champagne-Jorgensen, K. P. & Siddiqui, T. J. Low-Density Primary Hippocampal Neuron Culture. *J. Vis. Exp.* (2017) doi:10.3791/55000.
21. Asano, S. *et al.* Proteasomes. A molecular census of 26S proteasomes in intact neurons. *Science* **347**, 439–442 (2015).

22. Lavoie-Cardinal, F. *et al.* Neuronal activity remodels the F-actin based submembrane lattice in dendrites but not axons of hippocampal neurons. *Sci Rep* **10**, 11960 (2020).
23. Wiesner, T. *et al.* Activity-Dependent Remodeling of Synaptic Protein Organization Revealed by High Throughput Analysis of STED Nanoscopy Images. *Front Neural Circuits* **14**, 57 (2020).
24. Targa Dias Anastacio, H., Matosin, N. & Ooi, L. Neuronal hyperexcitability in Alzheimer's disease: what are the drivers behind this aberrant phenotype? *Transl. Psychiatry* **12**, 1–14 (2022).
25. Roselli, F. & Caroni, P. From Intrinsic Firing Properties to Selective Neuronal Vulnerability in Neurodegenerative Diseases. *Neuron* **85**, 901–910 (2015).
26. Berger, C. *et al.* Plasma FIB milling for the determination of structures in situ. *Nat Commun* **14**, 629 (2023).
27. Fedry, J. *et al.* Visualization of translation reorganization upon persistent ribosome collision stress in mammalian cells. *Mol. Cell* **84**, 1078–1089.e4 (2024).
28. Bean, B. P. The action potential in mammalian central neurons. *Nature Reviews Neuroscience* **8**, 451–465 (2007).
29. Gonzalez Sabater, V., Rigby, M. & Burrone, J. Voltage-Gated Potassium Channels Ensure Action Potential Shape Fidelity in Distal Axons. *J Neurosci* **41**, 5372–5385 (2021).
30. Lee, S. Y. *et al.* Cell-class-specific electric field entrainment of neural activity. *Neuron* **112**, 2659–2660 (2024).
31. Xing, H. *et al.* Translation dynamics in human cells visualized at high resolution reveal cancer drug action. *Science* **381**, 70–75 (2023).

REVIEWER COMMENTS

Reviewer #1 (Remarks to the Author):

The revised manuscript places a stronger emphasis on the technical advancements for establishing correlations between the electrophysiological properties and molecular structures of neurons. Additionally, the authors have improved the presentation and interpretation of the biological observations. I have no additional concerns.

Reviewer #2 (Remarks to the Author):

I am happy with the changes made to the manuscript.

Reviewer #3 (Remarks to the Author):

In this revised manuscript, Jung et al. have reorganized the text to provide a more comprehensive description of the CoVET technique. The authors provide more details in their specialized neuron culture, voltage imaging and in situ cryo-electron tomography (cryo-ET) approaches. The integration of electrophysiology with ultrastructural imaging represents a promising step forward and could serve as a valuable resource for future users and applications.

The authors report variations in ribosome states among neurons with differing excitability; however, the correlations between electrophysiological features and ribosome states are weak. Considering the low throughput of current cryo-ET, it would be valuable for the authors to discuss how this technique can be applied to address biological questions, or how the throughput could be increased.

We sincerely thank the reviewer for the thoughtful and constructive feedback. We are especially grateful that the reviewer understood and highlighted the key contributions of our work in integrating electrophysiology with in situ structural analysis. We also included current limitations in cryo-ET throughput and further biological application in the Discussion section briefly.

DISCUSSION:

“Moreover, overcoming current limitations in cryo-ET throughput through advances in sample preparation and data processing, such as high-throughput lamellae fabrication and AI based data processing, will enable CoVET to more precisely resolve the link between molecular and electrophysiological phenotype. We believe that CoVET, in combination with these technical developments, will play a pivotal role in uncovering diverse molecular mechanisms underlying neuronal responses, including protein translation and synaptic transmission.”

Major:

QIII-4-1. The width of individual action potential is typically a few ms, whereas the interspike intervals are typically much longer. Given that the duration of electrical field stimulation is only 1 ms, it is unlikely to induce multiple spikes within a 10 ms window.

I suspect that the observed variation in peak amplitude may be influenced by the low frame rate of the imaging system. A 10 ms sampling interval may fully capture narrow spikes but only partially capture wider ones, leading to inconsistent peak measurements. I recommend that the authors consider using the area under the curve (AUC) of each spike as a more robust measure of spike amplitude, especially given the low temporal resolution of their voltage imaging.

RIII-4-1. We sincerely thank the reviewer for the thoughtful and technically insightful comment. We agree that at a 10 ms sampling interval, voltage imaging may not fully capture the fine dynamics of individual action potentials, especially given their narrow width (~2 ms). Your suggestion to use area under the curve (AUC) as a more robust measure is well taken.

To carefully address this point, we performed high-speed voltage imaging at 2 kHz, which confirmed that electric field stimulation in our setup can indeed elicit multiple spikes within a 10 ms window (Fig. RIII-4-1a). These spikes are likely not solely direct responses to stimulation, but rather include recurrent network activity with varying synaptic delays, contributing to their temporal dispersion. As a result, the peak amplitude measured in our recordings reflects a temporally integrated neuronal response, and we consider it inherently captures aspects of the area under the curve (AUC).

Nonetheless, in response to your comment, we conducted an additional clustering analysis using AUC instead of peak amplitude. This reanalysis led to only a minor shift in clustering outcome, with 23 out of 569 neurons (~4%) changing their cluster identity (Fig. RIII-4-1b). This suggests that while the clustering is relatively robust to the choice of parameters, AUC and peak amplitude are not fully interchangeable and may highlight subtly different aspects of neuronal excitability.

We fully agree that the optimal parameter should depend on the imaging setup and biological question. Accordingly, we have now emphasized in the Discussion that the CoVET platform is inherently flexible and open to alternative clustering metrics:

DISCUSSION

“The choice of clustering parameters is not fixed but can be tailored to the experimental context. While we used peak amplitude in the present study, alternative metrics such as area under the curve (AUC) may offer complementary insights, particularly under different stimulation protocols or imaging regimes.”

Fig. RIII-4-1. a. 2kHz imaging with 1ms width electric field stimulation. **b.** PCA analysis was performed using peak value, decay parameter, and reproducibility. Clusters that change when using AUC instead of peak value are indicated by yellow circles.

QIII-4-3. The low-magnification images (Fig. 4a & b) suggest that neurons are likely interconnected via synapses. While spontaneous synaptic activity may be minimal in low-density cultures, synaptic modulation could become significant when neurons are actively spiking under electric field stimulation. As a result, the absence of synaptic blockers is still concerning. Synaptic inputs from neighboring neurons could disturb the interpretation of intrinsic properties, such as reproductivity, by activity dependent modulation from excited neighboring neurons.

Given the low throughput of cryo-ET and the relatively weak correlation between electrophysiological features and ribosome states, capturing broad phenotypes that are a mixture of intrinsic and extrinsic factors (authors use “functional output state of each neuron within its microenvironment”, which is confusing) makes it more difficult to have clear biological conclusions. As a methods-focused manuscript, the authors should emphasize the importance of experimental design and controls for future users applying this technique.

RIII-4-3: We thank the reviewer for the insightful comment and fully agree that synaptic connectivity can significantly influence neuronal responses, particularly under external stimulation. As also noted in RIII-4-1, our high-speed voltage imaging revealed that electric field stimulation often elicited multiple spikes within a 10 ms window, likely reflecting recurrent synaptic inputs in addition to direct excitability. This observation suggests that the functional readouts in our current setup represent a composite of intrinsic properties and network-mediated modulation.

We agree that the absence of synaptic blockers remains a valid concern when interpreting single-neuron electrophysiological phenotypes. In particular, synaptic inputs from neighboring neurons could affect parameters such as spike reproducibility and complicate the attribution of observed functional heterogeneity solely to intrinsic properties.

In response to the reviewer’s suggestion, we have now added a corresponding statement in the Discussion to guide future users:

DISCUSSION

“Moreover, neuronal maturity and synaptic connectivity among the neurons has been shown to strongly influence electrophysiological properties^{37,41} Variations in maturity and connectivity across the population further broadens the distribution of neuronal responses, resulting from both intrinsic excitability and synaptic input from neighboring neurons.”

“In relation to neuronal networks, future studies may benefit from combining CoVET analysis with synaptic blockers to isolate cell-autonomous activity, thereby enabling more accurate assessment of intrinsic neuronal responses independent of neuronal networks. Such isolation could also help disentangle intrinsic molecular features from network-driven variability, providing more precise insights into single-cell properties.”

Furthermore, we have revised the originally ambiguous phrase “functional output state of each neuron within its microenvironment” to clarify that our analysis captures a mixture of intrinsic excitability and extrinsic synaptic influence.

QIII-3. The geometry of the e-field setup is still confusing. Fig. 1.III suggests that the neurons are on the opposite side of the grid from the field-stim electrodes. Is this true? Fig. 3b is ambiguous about where the cells are. Please make clear whether the cells are on the top or bottom side of the grid in the cross-sectional view.

RIII-3. We fully agree that our original figure lacked clarity. The neurons are located on the side opposite to the electrodes, but this was not accurately conveyed in our illustration. Accordingly, we have revised the cross-sectional schematic to more clearly represent the experimental configuration, as you suggested (Fig. RIII-3).

Fig.RIII-3. Schematic illustration of grid holder with electric field stimulator

QIII-3-1. The cross-sectional simulation in Fig. RI-1-i.b shows a big electric field on one side of the grid, and much smaller on the other side, as one would expect from grid shielding. Unclear which side of the grid the map in panel c (which is the only thing shown in the manuscript) corresponds to. The authors need to characterize the variability in the electric field strength across the locations where the neurons are measured. There is big variation in E-field distribution based on the simulation results. The center of grids is around the 10V/m while the edge can achieve 1×10^4 V/m. Does this large difference in electric field dominate the variance observed for neural activity? The authors should provide some spatial maps showing the distribution of key electrophysiological features, including cluster types, peak value, decay parameter, reproducibility and spike numbers as a function of location on the grid.

RIII-3-1. We thank the reviewer for raising this important point. We agree that the large variation in electric field strength across the grid, as revealed by our simulation, could potentially contribute to spatial variability in neuronal responses. To directly address this concern, we performed spatial mapping of electrophysiological features—including peak amplitude, decay parameter, reproducibility, and spike number—across the entire grid.

Specifically, we grouped neurons into “central” ($-87\ \mu\text{m} < x < +87\ \mu\text{m}$, along the axis orthogonal to the electrodes) and “peripheral” populations, and compared their response features using unpaired t-tests. As shown in Fig. RIII-3-1 and Supplementary Fig. 3, we did not observe significant differences between the two regions across all major parameters. This suggests that the spatial heterogeneity in the electric field does not dominate the variance observed in neuronal activity under our experimental conditions. We interpret this apparent homogenization as a result of two contributing factors. First, the processes of mature hippocampal neurons typically extend over $\sim 500\ \mu\text{m}^1$, allowing each neuron to effectively integrate stimuli over a wide area, thereby averaging local E-field variations. Second, recurrent network activity among cultured neurons may further contribute to spatial smoothing of the input, via synaptic interactions and circuit-level integration.

In response to the reviewer’s suggestion, we have now incorporated the spatial analysis into both the *Methods* and *Results* sections of the revised manuscript, along with additional data presented in the *Supplementary Figures*.

METHODS:

“To identify the effect of an variability in electric field strength on neurons’ activity pattern, we defined neurons with x coordinate bigger than $-87\ \mu\text{m}$ and smaller than $+87\ \mu\text{m}$ as center neurons. Rest of the neurons were defined as peripheral neurons. (x axis was orthogonal to the electrodes). Peak height, decay parameter and reproducibility were compared between two groups using unpaired t-test.”

RESULTS:

“To further validate our experimental design in the context of local electric field variations along the Z axis (Supplementary Fig. 1a, b), we evaluated neuronal responses and clustering patterns based on their somatic coordinates. This analysis revealed no significant differences in response characteristics between neurons located in the central versus peripheral regions, nor among distinct neuronal clusters (Supplementary Fig. 3a-e), which supported the validity of our experimental scheme. These observations suggest that dendritic spatial integration may buffer local differences in field strength, leading to relatively uniform functional output across the network.”

Fig. RIII-3-1. Neuronal responses according to the location on the grids across 4 different grids. **a.** Comparison of neuronal responses in each parameter between neurons located at the center and periphery of the grids is shown as a violin plot. Dashed lines indicate quartile points. Unpaired t-test was performed to assess statistical significance. ns indicated

statistically non significant. **b-e**. Spatial mapping of neuronal responses **b**. Cluster, **c**. Peak value, **d**. Decay parameter **e**. Reproducibility is represented.

QIII-3-2. Also, the simulations lack information about the vectorial aspect of the electric field. Since the neurons are extended in the x-y plane, but comparatively flat in the z direction, in-plane vs. out-of-plane electric fields will have very different effects on the cells. Please separately show the in-plane and out-of-plane components. If, as I suspect, most of the electric field is in the z-direction, then a much more homogeneous electric field could be obtained by putting a single electrode directly above the sample (or a mesh, to permit light to go through), and applying the pulses between the sample chuck and the topside electrode, i.e. setting up a parallel-plate capacitor arrangement. The near-null in the electric field along the midline of the sample is consistent with my interpretation. An alternative to redesigning the sample chamber is just to connect the two Pt wires together, and to apply the voltage pulse between these two wires and the sample chuck.

RIII-3-2. We thank the reviewer for the constructive suggestion regarding the vectorial characteristics of the electric field. As recommended, we analyzed the electric field distribution by separating it into in-plane (x-y) and out-of-plane (z) components based on finite element simulations. As shown in Fig. RIII-3-2a,b, the in-plane electric field was relatively uniform across the grid, whereas the out-of-plane (z-axis) component exhibited substantial spatial heterogeneity, particularly near the edges of the grid.

This observation supports the reviewer's interpretation that the dominant component of the electric field in our setup is aligned along the z-axis, and further explains the field null along the central plane of the grid. While we did not observe significant spatial differences in neuronal responses (see RIII-3-1), we agree that the observed heterogeneity in the out-of-plane electric field could become more impactful under different stimulation regimes or culture conditions. To this end, future studies should systematically explore how variations in field geometry, frequency, and neuronal maturation state influence stimulation outcomes in CoVET.

Accordingly, we addressed the variation of the electric field in the Results section and incorporated your suggested improvements regarding our electric field stimulation configuration into the revised Discussion section, as follows:

RESULTS

"To further validate our experimental design in the context of local electric field variations along the Z axis (Supplementary Fig.1a, b), we evaluated neuronal responses and clustering patterns based on their somatic coordinates. This analysis revealed no significant differences in response characteristics between neurons located in the central versus peripheral regions, nor among distinct neuronal clusters (Supplementary Fig. 3a-e), which supported the validity of our experimental scheme. These observations suggest that dendritic spatial integration may buffer local differences in field strength, leading to relatively uniform functional output across the network."

DISCUSSION

"Regarding the electric field stimulation, in our current configuration, lateral platinum electrodes were used to apply electric field stimulation across the neuronal grid (Fig. 3a, b). Simulation and vector decomposition analyses revealed that this setup generates a non-

uniform electric field, particularly along the out-of-plane (z) axis (Supplementary Fig. 1). Although we did not observe correlated neuronal responses under our stimulation protocol (Supplementary Fig. 3), such field variability could become more influential under different experimental conditions, such as higher-frequency stimulation, altered culture densities, or varying neuronal maturation states. Systematic validation of stimulation geometries under diverse experimental contexts will be important for optimizing CoVET and ensuring robust interpretation of functional phenotypes. Furthermore, the choice of clustering parameters is not fixed but can be tailored to the experimental context. While we used peak amplitude in the present study, alternative metrics such as area under the curve (AUC) may offer complementary insights, particularly under different stimulation protocols or imaging regimes.”

Fig. RIII-3-2. 3D electric field simulation. **a.** Vector of the electric field shown in various views. Electrodes are positioned at x-axis. **b.** Magnitude of applied electric field along x, y and z axis.

Other:

QIII-4. Fig. 3f, middle: Why does “Decay parameter” have units of dF/F ?

RIII-4. We thank the reviewer for pointing out the issue regarding the unit and terminology of the “Decay parameter.” We now recognize that we had misunderstood the implications of the term, which typically implies a temporal dynamic or rate. In our analysis, however, this value was calculated as the difference between the peak $\Delta F/F$ and the average $\Delta F/F$ across frames 2 to 6 after stimulation—effectively a scalar quantity without a time component. We included this aspect of the Decay parameter in the Methods section.

METHODS:

“The decay parameter was calculated by subtracting the average of the 2th to 6th frames from the peak value, reflecting the extent of decay as a scalar value.”

We also acknowledge that the physiological interpretation of such parameters may require further validation. We anticipate that future studies incorporating higher temporal resolution and complementary measurements will help establish the robustness and biological significance of these metrics.

Minor:

QIII-5. QIII-1-2 & Line 95 in manuscript: The use of "cells/mL" to describe the seeding density of a 2D primary neuron culture is confusing. (Is that the concentration of resuspended neurons?) Authors also used cells/cm² when demonstrating their low-density culture, which is inconsistent with literature standards.

RIII-5. We thank the reviewer for pointing out this inconsistency. Cells/ml meant the concentration of resuspended neurons, as you pointed out. We agree that this unit is not appropriate for describing 2D culture density and may cause confusion. To improve clarity and consistency with standard practice in the literature, we have revised the text to uniformly express all cell densities in units of “cells/cm²” as follows:

RESULTS:

“Conventional hippocampal primary cultures required a seeding of ~120,000 cells/cm², resulting in a high-density cell population²⁴.”

QIII-6. Abstract and line 87: The authors refer to ribosomes having different “contextual information”. Unclear what this means. Please rephrase to be more clear and specific.

RIII-6. Thank you for your suggestion. We have revised the abstracts accordingly for the clearance of expression

ABSTRACT

“We analyzed the translational landscapes of individual neurons, and found distinct structural characteristics and spatial networks among ribosomes from different electrophysiological clusters.”

QIII-7. Line 635: “Each column is shown with mean \pm s.e.m.” It took me a while to figure out that the dashed horizontal lines are supposed to represent these values, rather than individual data points. Consider showing mean and variability with a box. Also, it looks more like s.d. to me than s.e.m. Please check.

RIII-7. Thank you for your careful reading. That was our mistake. In the violin plot used to illustrate batch variability across the grids, the dashed lines represented the quartiles. Additionally, the statement “Each column is shown with mean \pm s.e.m” should have been included in the legend for Fig. 3I. In this figure, the s.e.m. was used to visualize the variability. We have revised our figure legends accordingly.

Fig. 3 legends

“f, Batch-to-batch variability in the response parameters is plotted as a violin plot. Dashed lines indicated quartiles.

*i, Parameters used for HCA from each cluster are shown. (i) Peak values, (ii) decay parameter, and (iii) reproducibility of action potentials from neuronal responses to electric field stimulation were measured in each cluster and plotted as bar plots. Each column is shown with mean \pm s.e.m. One-way analysis of variance (ANOVA) followed by Tukey’s post hoc test for multiple comparisons was performed to assess statistical significance. * $P < 0.05$, *** $P < 0.001$, **** $P < 0.0001$.”*

QIII-8. Line 397: still refers to AUC instead of “decay parameter”.

RIII-8. Thank you for your careful reading. We have revised our manuscript accordingly.

METHODS

“Using the dF/F dataset from four grids, we quantified the response peak value, decay parameter, and reproducibility of action potentials over a time window of 20 ms before and 480 ms after electric field stimulation (-2 to +48 frames).”

QIII-9. Line 401: The definition of reproducibility doesn’t make sense. Please check this sentence carefully, some words are missing.

RIII-9. Thank you for your careful reading. We have revised accordingly.

METHODS

“To assess reproducibility, we defined it as the proportion of evoked action potentials out of ten repeated stimulations. For this analysis, action potentials were defined as dF/F peak values exceeding 3.5% and a peak signal-to-noise ratio greater than 8 dB. ”

References

1. Park, H.-A., Licznerski, P., Alavian, K. N., Shanabrough, M. & Jonas, E. A. Bcl-xL is necessary for neurite outgrowth in hippocampal neurons. *Antioxid Redox Signal* **22**, 93–108 (2015).